# A proteomic chronology of gene expression through the cell cycle in human myeloid leukemia cells

**Tony Ly[1], Yasmeen Ahmad[1], Adam Shlien[2], Dominique Soroka[3], Allie Mills[3], Michael J Emanuele[3], Michael R Stratton[2], Angus I Lamond[1]\***

[1]Centre for Gene Regulation and Expression, College of Life Sciences, University of Dundee, Dundee, United Kingdom; [2]Wellcome Trust Genome Campus, Wellcome Trust Sanger Institute, Cambridge, United Kingdom; [3]Lineberger Comprehensive Cancer Center, University of North Carolina at Chapel Hill, Chapel Hill, United States

**Abstract** Technological advances have enabled the analysis of cellular protein and RNA levels with unprecedented depth and sensitivity, allowing for an unbiased re-evaluation of gene regulation during fundamental biological processes. Here, we have chronicled the dynamics of protein and mRNA expression levels across a minimally perturbed cell cycle in human myeloid leukemia cells using centrifugal elutriation combined with mass spectrometry-based proteomics and RNA-Seq, avoiding artificial synchronization procedures. We identify myeloid-specific gene expression and variations in protein abundance, isoform expression and phosphorylation at different cell cycle stages. We dissect the relationship between protein and mRNA levels for both bulk gene expression and for over ~6000 genes individually across the cell cycle, revealing complex, gene-specific patterns. This data set, one of the deepest surveys to date of gene expression in human cells, is presented in an online, searchable database, the Encyclopedia of Proteome Dynamics (http://www.peptracker.com/epd/).

**\*For correspondence:**
a.i.lamond@dundee.ac.uk

**Competing interests:** The authors declare that no competing interests exist.

## Introduction

Recent technological advances in both mass spectrometry and nucleic acid sequencing have created new high throughput methods for the quantitative measurement of protein (*Lamond et al., 2012*; *Mann et al., 2013*) and RNA levels (*Nagalakshmi et al., 2008*; *Wilhelm et al., 2008*). This in turn has allowed studies, in both model organisms and human cells, which seek to document global proteomes and transcriptomes. For example, several laboratories have performed in-depth proteomic profiling of different human cell lines and independently concluded that the minimum number of proteins expressed in each cell line is on the order of 10,000 (*Beck et al., 2011*; *Nagaraj et al., 2011*). In a more recent study, the proteomes of 11 cell lines were profiled to a similar depth (*Geiger et al., 2012*). These data enable comparison of protein expression levels between cells that differ in tissue type, developmental origin, and mode of in vitro culture. At the transcriptome level, microarray analysis and RNA-Seq have been used to document global mRNA expression across extensive panels of human cells and tissues (*Ramaswamy et al., 2001*; *Su et al., 2002*), and to examine features of RNA regulation, such as alternative splicing (*Braunschweig et al., 2013*).

Although global quantification of mRNA levels is often more convenient, several studies have reported that for human cells and model organisms RNA levels alone are not uniformly predictive of protein levels (*Gygi et al., 1999*; *Vogel et al., 2010*; *Maier et al., 2011*; *Nagaraj et al., 2011*; *Schwanhausser et al., 2011*). The relationship between protein and mRNA abundance remains a topic of debate and the studies above highlight the challenges of relying on analysis of mRNA alone

**eLife digest** Cells are complex environments: at any one time, thousands of different genes act as molecular templates to produce messenger RNA (mRNA) molecules, which themselves are templates used to produce proteins. However, not all genes are active at all times inside all cells: as cells grow and divide as part of the cell division cycle, genes are switched on and off on a regular basis. Similarly, the patterns of mRNA and protein production are different in, say, immune and skin cells.

In recent years, the tools available for detecting mRNA molecules and proteins have become more powerful, allowing researchers to move beyond just measuring the total amounts of mRNA and protein in the cell to now measuring individual amounts of specific mRNA and protein molecules encoded by specific genes. However, it has been a challenge to make these measurements at different stages of the cell cycle. Most of the methods used to do this have involved artificially 'arresting' the cell cycle, which can lead to side effects that are difficult to account for.

Ly et al. have now overcome these problems using a combination of three methods to measure the levels of mRNA and protein molecules associated with over 6000 genes in human cancer cells derived from myeloid leukemia. Exploiting the fact that cells change size during the cell cycle, Ly et al. used a centrifugation technique to separate cells based on their size and, therefore, the stage of the cell cycle they were at, thus avoiding the need to arrest the cell cycle. An approach called RNA-Seq was then employed to measure the levels of the different mRNA molecules in the cells, and a device called a mass spectrometer was used to identify and measure the levels of many different proteins.

In addition to being able to follow the level of mRNA and protein production for a large number of genes throughout the cell division cycle, while also obtaining detailed information about how many of the proteins are modified, Ly et al. discovered that—contrary to expectations—low numbers of mRNA molecules were sometimes associated with high numbers of the corresponding protein, and vice versa. This work provides a better understanding of the complex relationship between the levels of an mRNA and its corresponding protein product, and also demonstrates how it may be possible to detect subtle but important differences between cell types and disease states, including different types of cancer.

to measure gene expression at the protein level. High throughput methods have also been used to analyze changes in gene expression in response to specific events, including cell cycle progression. Our objective in this study was to capitalize on these technological advances to provide an in-depth characterization of gene expression in human cells, including cell cycle-associated changes in the proteome and transcriptome. We thus address the key question of how different layers of gene expression affect corresponding levels of protein and mRNA in a biologically important, dynamic system.

The mitotic cell cycle is a conserved and highly regulated process in all eukaryotes, which has been categorized into four consecutive phases, that is, Gap 1 (G1), Synthesis (S), Gap 2 (G2) and Mitosis (M). Regulation of the cell cycle is important for controlling cell growth and proliferation and for coordinating the timing of major cellular events, such as DNA replication and cell division (*Hunter and Pines 1994*). Regulatory pathways and checkpoints allow cells to respond quickly to DNA damage and other forms of stress that require cell cycle arrest to prevent uncontrolled cell division (*Hartwell and Kastan, 1994*; *King et al., 1994*; *Elledge, 1996*; *Pines, 1999*). Many signaling mechanisms also impact on the control of cell cycle to allow cells to grow and divide in response to both developmental and environmental cues. Misregulation of the cell cycle machinery can lead to inappropriate cell proliferation, as often seen in neoplastic disease.

There are significant technical challenges involved in the analysis of cell cycle-dependent regulation of gene expression. Most cell cycle analyzes make use of cell synchronization methods to enrich populations of cells at specific cell cycle stages in sufficient quantities for biochemical characterization. For example, multiple strategies have been used to characterize levels of mRNA expression at different cell cycle stages in the budding yeast, *Saccharomyces cerevisiae*, including conditional knockdown of cell cycle regulators, withdrawal of growth factors, use of chemical inhibitors and physical size separation

by centrifugal elutriation (*Cho et al., 1998*; *Spellman et al., 1998*). Large-scale transcriptome analyzes have also been performed in mammalian cells, particularly in HeLa cells, to compare mRNA expression levels across the cell cycle (*Cho et al., 2001*; *Whitfield et al., 2002*). More recently, several groups have also examined cell cycle variation in the mammalian proteome and phosphoproteome in cell line models (*Ohta et al., 2010*; *Olsen et al., 2010*; *Pagliuca et al., 2011*; *Lane et al., 2013*). The established methods to achieve highly synchronized mammalian cells usually involve arresting cells, either by inducible genetic depletion of factors needed to drive cell progression, or by drug treatments that either activate checkpoints, block major metabolic pathways, or else disrupt the mitotic spindle. Inhibiting or depleting essential factors and activities needed for proper cell cycle progression, however, may in turn cause side effects that alter gene expression independent from direct, cell cycle-based regulation (*Cooper et al., 2007*).

An additional challenge is that several studies have suggested that there may be tissue-specific plasticity in cell cycle regulation (*Pagano and Jackson, 2004*). For example, studies in mice with genetic deletions of D-type cyclins have shown that the hematopoietic system is the only tissue that requires D-type cyclins for cell proliferation (*Kozar et al., 2004*). In contrast to epithelial tumor cell lines, large-scale studies examining protein expression in hematopoietic cells are sparse, with the Jurkat-T and K562 cell lines being the only immune cell lines comprehensively profiled (*Geiger et al., 2012*). To the best of our knowledge, there have been no previous large-scale studies on myeloid cell cycle gene expression.

We have addressed these challenges by undertaking a high-resolution proteomic analysis of cell cycle gene expression in human NB4 cells. These cells are derived from the myeloid lineage and have been widely used as a model system for studying acute promyelocytic leukemia and myeloid biology (*Drexler et al., 1995*), due to their 'undifferentiated' promyelocyte state (*Grisolano et al., 1997*; *He et al., 1997*; *Zhu et al., 2005*). The data characterize the proteome of NB4 cells to a depth of over 10,000 proteins, with high average sequence coverage, including analysis of isoform expression and post-translational phosphorylation. We analyze cell cycle-regulated gene expression in NB4 cells at both the protein and mRNA level, using counterflow centrifugal elutriation (*Banfalvi, 2008*) combined with high-throughput, label free mass spectrometry-based proteomics and RNA-Seq. We identify subsets of genes encoding proteins whose abundance is cell cycle regulated, including novel factors, isoforms, and phosphorylation sites. All of the resulting data have been incorporated into the Encyclopedia of Proteome Dynamics (http://peptracker.com/epd/), an online, free to access, searchable database.

## Results

### Experimental design

Our goal was to obtain deep proteome coverage that would document global gene expression in the human myeloid cell lineage and to combine this with an unbiased, quantitative chronology of changes in gene expression across the mitotic cell cycle, with minimal perturbation to cellular physiology. To achieve this, we analyzed the human NB4 promyelocytic leukemia cell line using a strategy that combines centrifugal elutriation (*Banfalvi, 2008*), a physical method of enriching cell populations at different cell cycle stages, with high throughput analyzes of both protein and poly(A)+ mRNA levels. This strategy is outlined schematically in *Figure 1*. First, six elutriated fractions were collected from an unsynchronized population of NB4 cells grown in normal, label free medium. Second, to obtain deep proteomic information with high peptide and protein coverage, extracted proteins from each elutriated fraction were digested with trypsin and/or LysC and further fractionated by offline peptide chromatography prior to mass spectrometric analysis. As described below, this allowed quantitative profiling of the NB4 proteome to a depth of >10,000 proteins, with high mean sequence coverage (~38%). Third, poly(A) + RNA was also isolated from each of the same elutriated NB4 cell fractions that were used to isolate proteins and analyzed by an RNA-Seq transcriptomics workflow (*Figure 1*). This allowed us to quantitate transcripts from 12,078 protein coding genes, including 9667 genes whose mRNA expression was measured separately in each of the three pooled, elutriated fractions and in asynchronous cells.

### Enrichment of cell cycle phases by centrifugal elutriation

Cells from an asynchronous, unfractionated NB4 cell culture and from each of the six elutriated fractions were analyzed by both flow cytometry and protein blotting to characterize their cell cycle profiles (*Figure 2*). First, cells in the total unfractionated NB4 population and in each of the six elutriated fractions were stained with propidium iodide (PI) and analyzed by flow cytometry. Histograms showing the results

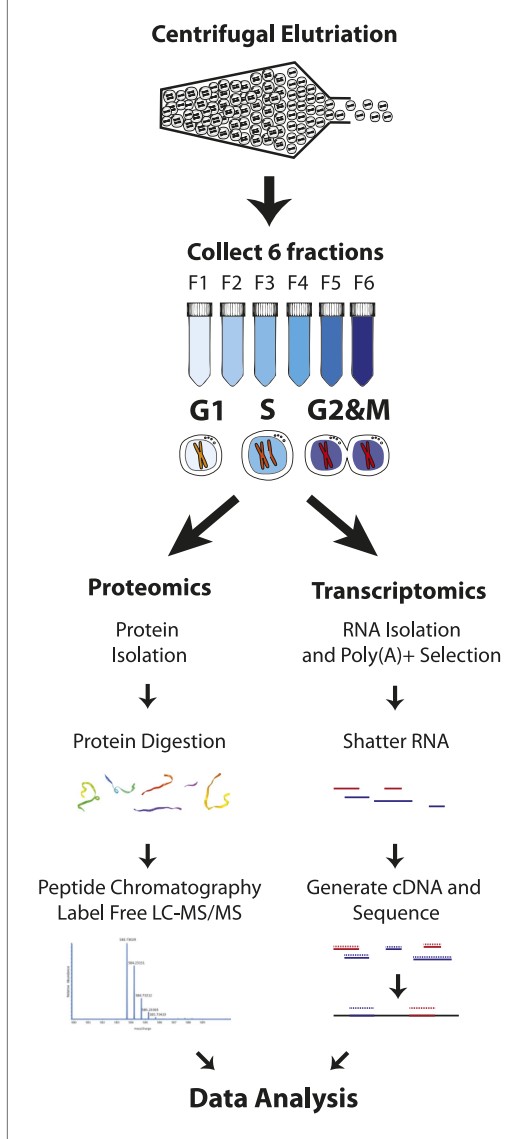

**Figure 1**. Experimental workflow. NB4 cells were harvested and fractionated by cell size using centrifugal elutriation. Six fractions were collected and processed separately for transcriptomics and proteomics. For proteomics, cells were lysed and digested with either Lys-C or Lys-C/trypsin. Peptides were then separated by two orthogonal modes of chromatography prior to analysis using an Orbitrap mass spectrometer. Data normalization, peak picking, database searching, peptide and protein identification were performed using the MaxQuant software suite. For transcriptomics, cells from the six fractions were pooled into three (G1, S, and G2&M-enriched fractions). Total RNA was extracted, and subjected to poly(A)+ tail selection. Poly(A)+ transcripts were shattered, reverse transcribed to establish cDNA libraries, which were sequenced using Illumina paired-end sequencing technology. Reads were aligned to the human genome (build hg19) using TopHat, and then used for quantitative gene expression analysis of known protein coding genes using Cufflinks.

of PI fluorescence measurements confirm that there is a pronounced differential enrichment for populations of cells in G1–G2 and M phases across elutriated fractions 1–6 (*Figure 2A*). The proportion of cells in each cell cycle phase was calculated by fitting the data to the Watson model (*Watson et al., 1987*). Fractions F1 and F2 were enriched in G1 cells, F3 and four were enriched in S phase cells and F5 and F6 were enriched in G2 and M phase cells (G2&M).

Second, we used immunoblotting to compare the levels of selected marker proteins, whose expression across the cell cycle is known, with the enrichment for each cell cycle stage detected by flow cytometry (*Figure 2B*). Equal amounts of total NB4 proteins from either the asynchronous cell population or from each elutriated fraction were separated by SDS-PAGE, electroblotted onto nitrocellulose and marker proteins detected using specific antibodies, as described in the 'Materials and methods'. This shows that, as expected, GAPDH is expressed at similar levels across all fractions, while cyclin E, which is a marker for the G1/S transition, has maximal expression in fractions F1 and F2. In contrast, markers for G2 and M phase cells, such as cyclin B1, aurora kinase B and phospho-Histone H3 (S10), all have maximal expression in the last fractions (F5 and F6). All of the protein blotting data are thus consistent with the profiles of cell cycle stage enrichment deduced from the flow cytometry analysis and confirm that the six elutriated fractions are differentially enriched for NB4 cells at different stages of progression through interphase and into mitosis.

We separately tested for viability of the collected NB4 cells post elutriation. Re-inoculation of the fractionated NB4 cells into tissue culture medium showed that the bulk of the cells survived the elutriation procedure with high viability and minimal damage and rapidly resumed growth when returned to culture, as judged by subsequent FACS analysis of the replated cells (*Figure 2C*, right). Elutriated NB4 cells were of similar size and granularity (as measured by forward and side scatter, *Figure 2C*, left), as a control NB4 cells that were not exposed to elutriation.

In summary, based on the combination of flow cytometry, immunoblot, and cell culture analyzes, we conclude that the elutriation strategy provides an effective physical method for fractionating unsynchronized populations of human immune NB4 cells into viable subpopulations of cells that are enriched in distinct cell cycle phases, with minimal perturbation to normal cell physiology and viability. Further, the elutriation methodology is highly reproducible, as documented below.

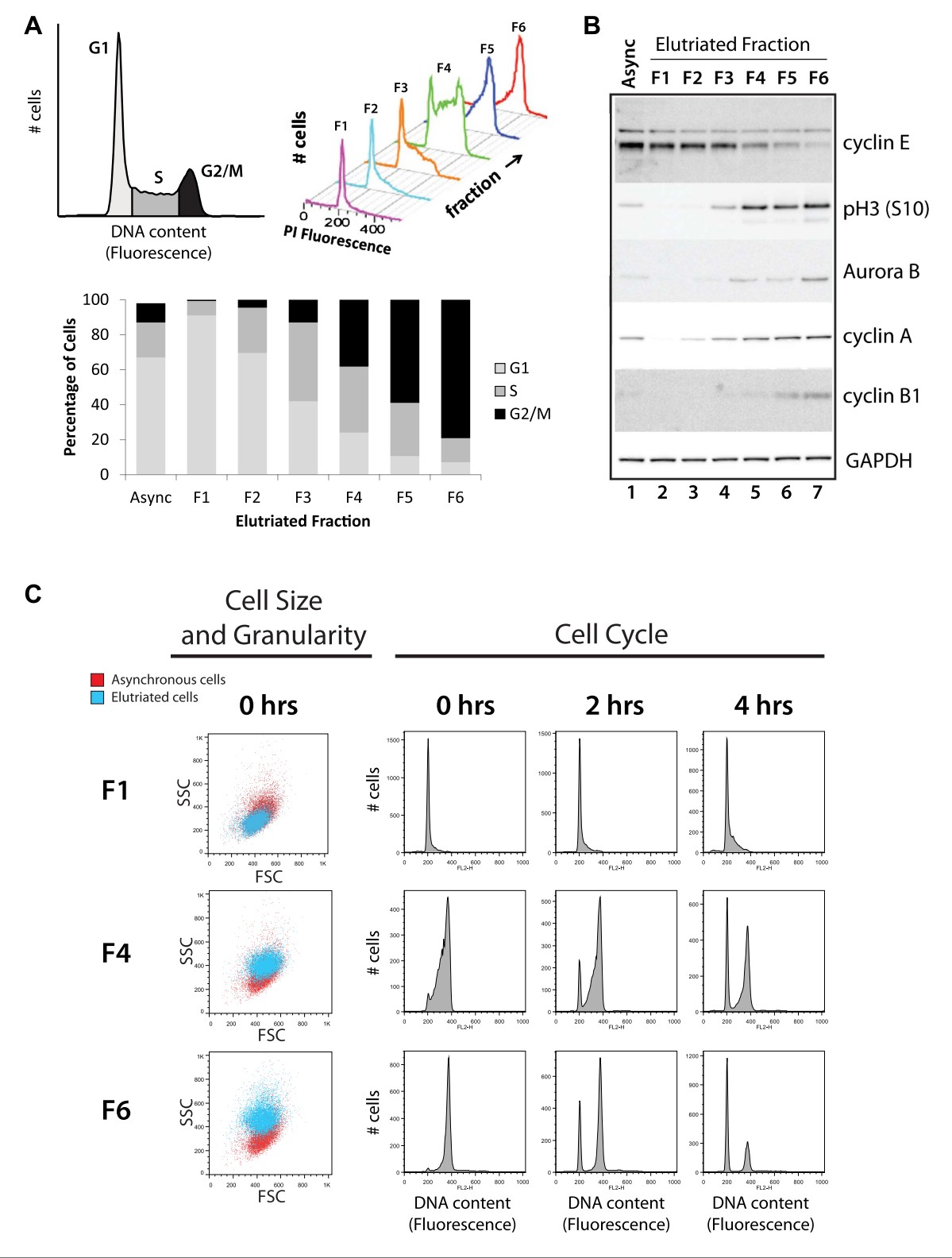

**Figure 2**. Validation of cell cycle enrichment by centrifugal elutriation. (**A**) Cells from asynchronous cells (top left) and each elutriation fraction (top right) were stained with a DNA-binding fluorescent dye and analyzed with flow cytometry. Proportions of cells in each cell cycle phase (bottom) were estimated using the Watson model. Fractions 1 and 2 (F1 and F2) are enriched in G1, fractions 3 and 4 (F3 and F4) are enriched in S, and fractions 5 and 6 (F5 and F6)

*Figure 2. Continued on next page*

*Figure 2. Continued*

are enriched in G2 and M phase (G2&M). (**B**) Immunoblot analyses of the protein lysates for known cell cycle phase-specific markers (cyclin E, phospho-Histone H3 S10, aurora kinase B, cyclin A, and cyclin B1) are consistent with previous literature and the enrichment profiles in (**A**). (**C**) Forward scatter and side scatter plots (first column) and cell cycle distributions (remaining columns) for three representative fractions post inoculation (F1, F4, and F6). The forward and side scatter plots for each elutriated fraction are shown in cyan, and are directly compared with the same plot for asynchronous cells, which is shown in red. Cell cycle distributions of the three fractions are measured directly after elutriation (0 hr), and 2 and 4 hr after inoculation into tissue culture medium. Note that the cell cycle distributions shown includes cells with <2N DNA content.

## Estimation of biological and technical variance

To evaluate the biological and technical reproducibility of the elutriation strategy, we performed three rounds of elutriation on NB4 cultures harvested on different days. Proteins were isolated from each elutriated fraction from each biological replicate. Samples were processed for mass spectrometry using a Single Shot workflow and analyzed in technical triplicate by reversed phase LC-MS/MS. Label free intensities were calculated from MS peptide-extracted ion chromatograms, as previously described (*Luber et al., 2010*). Comparison of the label free intensities within both the technical and biological replicates shows that the average Pearson correlation coefficients are greater than 0.97, indicating that both the elutriation-based cell cycle enrichment and the label free MS-based peptide quantitation methods are highly reproducible (*Figure 3—figure supplement 1*).

## Characterization of the global NB4 cell proteome

To achieve deep coverage of the total NB4 cell proteome, we used a proteomic workflow combining digestion with two proteases with extensive pre-fractionation of peptides prior to MS analysis. Thus, protein samples isolated from each elutriate from a single elutriation experiment were divided and digested with either Lys-C alone (Lys-C), or double digested with Lys-C and Trypsin (Trypsin-DD). The resulting peptides were separated by analytical, hydrophilic Strong Anion eXchange (hSAX) chromatography into 12 fractions (*Ritorto et al., 2013*). Each fraction was then analyzed by LC-MS/MS.

Using the proteomics workflow described above, over 150,000 peptides were identified (Dataset S1, entire file available at Dryad, *Ly et al., 2014*), corresponding to 10,929 unique protein groups, over 10,000 of which were detected with two or more peptides. These proteins represent expression of more than 9000 genes. As shown in *Figure 3A*, the quantitative data set includes proteins whose MS-measured extracted ion chromatogram intensities span eight orders of magnitude, which corresponds to at least four orders of magnitude in protein copy number (*Nagaraj et al., 2011*). A wide variety of biological functions are captured, reflected by the wide range of GO annotations, from low abundance proteins involved in transcription, to very high abundance proteins, such as histones and factors involved in ribosome subunit assembly (*Figure 3A*).

Given the wide range of protein expression levels, the bulk of protein abundance in NB4 cells results from the expression of a relatively small number of proteins, as is shown in a cumulative protein abundance plot (*Figure 3B*). Thus, half of the total protein molecules expressed correspond to only ~90 highly abundant proteins (0.9% of the proteins detected), which is similar to previous reports on other cell types (*Nagaraj et al., 2011*). In the other extreme, 10% of the total protein abundance reflects the expression of over 9000 different proteins, which highlights the prerequisite of detecting proteins across a wide range of abundance values for comprehensive proteome characterization.

We have generated here deep coverage of the NB4 proteome with a group of over 10,000 proteins identified that are each supported by an average of 15 separate peptides (*Supplementary file 1*). In total, 154,985 sequence-unique peptides were detected, of which 30.8% were identified by only Lys-C, and 50.5% by only Trypsin-DD, as shown in *Figure 3C*. These peptides map to a total of 1,976,427 unique amino acid locations in the UniProt Human Reference Proteome (*Figure 3D*), yielding on average 139 amino acids sequenced per protein. As shown, Lys-C and Trypsin-DD peptides map to largely complementary sequence regions (*Figure 3D*). Thus, the combined use of these two proteases significantly increases the overall sequence coverage. As shown in *Figure 3E*, use of complementary proteases improves the mean sequence coverage by over 8% compared with a single digestion method. This yields a combined mean sequence coverage of ~38% for over 10,000 proteins, providing one of the most detailed protein expression maps so far reported. Furthermore, using the double protease strategy, many proteins with low sequence coverage based on single protease digestion

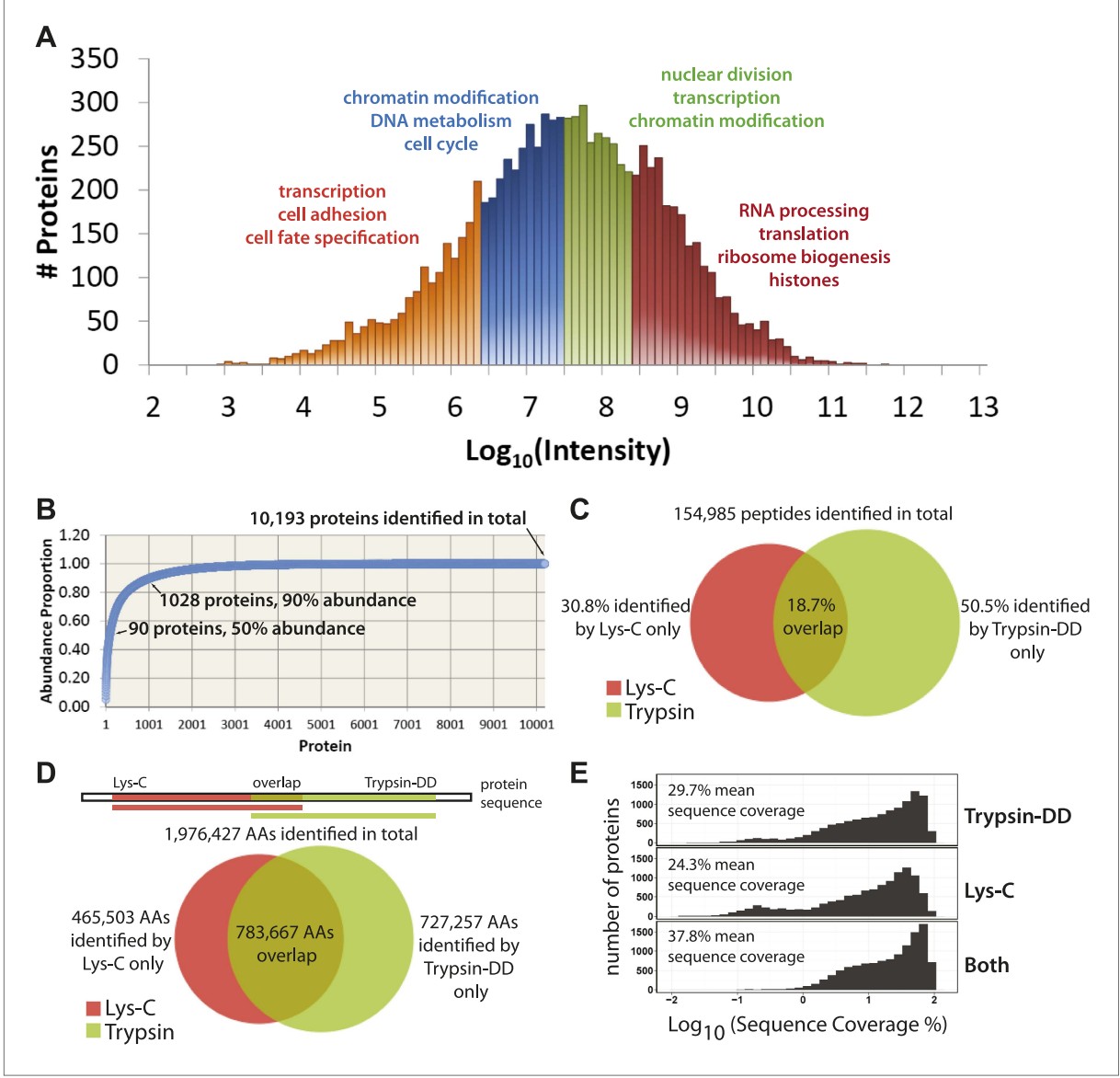

**Figure 3**. Quantitative, in-depth characterization of a myeloid leukemia proteome. (**A**) A histogram of log-transformed protein abundance (iBAQ-scaled protein intensities). Quartile regions are shown in different colors, and enriched gene ontology terms (p<0.01) are shown above each region. (**B**) A cumulative plot of protein abundance, as estimated using iBAQ-scaled intensities. In total, 10,193 proteins were identified with at least two supporting peptides per protein. Protein abundances follow an exponential increase, with 90 proteins (0.9%) constituting 50% of the bulk protein mass, and 1028 proteins (10%) constituting 90% of the bulk protein mass. The remaining protein identifications (9075 or 89.1%) comprise less than 10% of the bulk protein mass in NB4 cells. (**C** and **D**) Venn diagrams showing the total number of sequence-unique peptides (154,985) and amino acid coverage (1,976,427) split by digestion method. Lys-C increases the number of peptides identified by 44% relative to Trypsin-DD. Amino acid coverage was calculated by mapping sequences back to an assembled proteome. Over 30% of the amino acids detected using Lys-C digestion reside in sections of protein sequences that are complementary to Trypsin. In summary, complementary digestion methods substantially increase the overall sequence coverage, as shown in (**E**). Combining data from both methods boosts the mean sequence coverage to 37.8% with comprehensive proteome depth of over 10,000 proteins.

The following figure supplements are available for figure 3:

**Figure supplement 1**. Estimation of technical and biological variances among replicates indicates highly reproducible protein quantitation.

**Figure supplement 2**. Comparison of expected versus observed amino acid and gene ontology frequencies reveals no major detection bias in the proteomics data set.

(observed as a second, broad peak to the left in *Figure 3E*), are shifted to higher sequence coverage. High sequence coverage not only improves the accuracy of protein and isoform identification, it is also important for high resolution analysis of protein regulation during the cell cycle, as highlighted below.

We next evaluated the NB4 data set for possible detection bias, for example due either to differential efficiency of extracting distinct classes of protein in the lysates prepared, or to failure by MS to detect protein types featuring high levels of modification and/or unusual sequence combinations that are not cleaved efficiently into peptides. To address this, we compared the observed frequencies of GO terms and the amino acid proportions for all of the proteins detected in this NB4 cell data set with the corresponding predicted frequencies calculated from *in silico* translation of the entire human proteome (*Figure 3—figure supplement 2*). The Pearson correlation coefficients observed between these predicted and measured frequencies (r >0.98) indicate that the sampling of proteins in the NB4 data set is highly representative of the human proteome. While inevitably some expressed proteins have not been detected, particularly in the very low abundance range, we can effectively exclude that there is a major bias, either from under-sampling specific protein classes (e.g., membrane proteins), or from an absence of lower abundance proteins in general.

## Comparison of the NB4 proteome with other human cell line proteomes

Next, we compared this proteome analysis of NB4 cells, a human promyelocytic leukemia cell line that grows in suspension culture, with other recent examples of in depth proteomic analysis of different human cell lines, most of which are adherent tumor cell lines, of either fibroblast or epithelial origin. This meta-analysis included protein data from 14 cell line proteomes: 3 × HeLa, 2 × U2OS, A549, GAMG, HEK293, K562, LnCap, MCF7, RKO, HepG2, and Jurkat-T (*Lundberg et al., 2010*; *Beck et al., 2011*; *Nagaraj et al., 2011*; *Geiger et al., 2012*), which were consolidated and mapped to Ensembl Genes prior to comparison. The combined data set provides evidence of protein-level expression of over 11,000 human genes. Of these, a common set of ~3000 genes are identified by protein data from all these cell lines, defining a core, shared proteome (*Supplementary file 2*). Interestingly, the abundance values of proteins in this core proteome span the full abundance range of the entire NB4 proteome. This suggests that the core proteome is not simply reflecting a detection bias towards abundant proteins. The core proteome is enriched in proteins associated with RNA processing, translation, cell cycle, and DNA metabolic processes, which together highlight key biological processes required for cell proliferation. In contrast, analysis of cell type-specific proteomes highlight specialized biological functions that are associated with cell lineage and mode of culture, as will be discussed below.

Approximately, 10% of the expressed genes we detected in NB4 cells at the protein level are exclusive to this study and have not been reported in large-scale proteomic studies of other human cell lines (listed in *Supplementary file 2*). Interestingly, this NB4-specific pool is enriched in proteins that regulate cation flux in the cell, proteins involved in the innate immune response, zinc finger proteins and transcription factors (>200), including proteins known to be important to leukemic and immune cell biology, such as RARα, RXRβ, CEBPα, GFI-1 and PU.1 (*Zhu et al., 2001*; *Orkin and Zon, 2008*).

We next focused on comparing the NB4 proteome with the most recent study describing in detail protein expression in several human cell lines (*Geiger et al., 2012*), including the K562 and Jurkat-T cancer cell lines derived from the immune lineage (myeloid and lymphoid, respectively), that are the most related to NB4 (myeloid). The other two cell lines compared (HeLa and MCF7) are derived from epithelial tumors. Pairwise comparisons were performed to determine sets of genes that are uniquely detected in each cell line. Enriched gene ontology terms for each set are shown in *Figure 4A*. Comparison of these cell line-specific subproteomes reveals proteins with functions that highlight not only the differences in lineage, but also distinguish mode of culture, for example suspension vs adherent culture. For example, HeLa- and MCF7-specific sets are enriched in genes involved in cell adhesion, such as cadherins and integrins, whereas the Jurkat-T-specific set is enriched in genes involved in T-cell selection and activation, such as CD1, CD3, and CD4 (*Figure 4A*).

For three out of the four cell lines, pairwise comparisons reveal specific transcription factors that are enriched in the NB4-specific data set, as similarly found for the broader comparison described above. In contrast, comparison of the NB4 proteome with the proteome of the myeloid K562 cells reveals that many transcriptional regulators are shared between these two myeloid cell lines. Thus, among the 87 genes that express proteins in K562 and NB4, but which are not detected in MCF7 and HeLa (*Geiger et al., 2012*), 22 are either known or putative transcription factors including SP1 and JUN (*Friedman, 2002*), and five have been annotated with gene ontology terms associated with myeloid differentiation

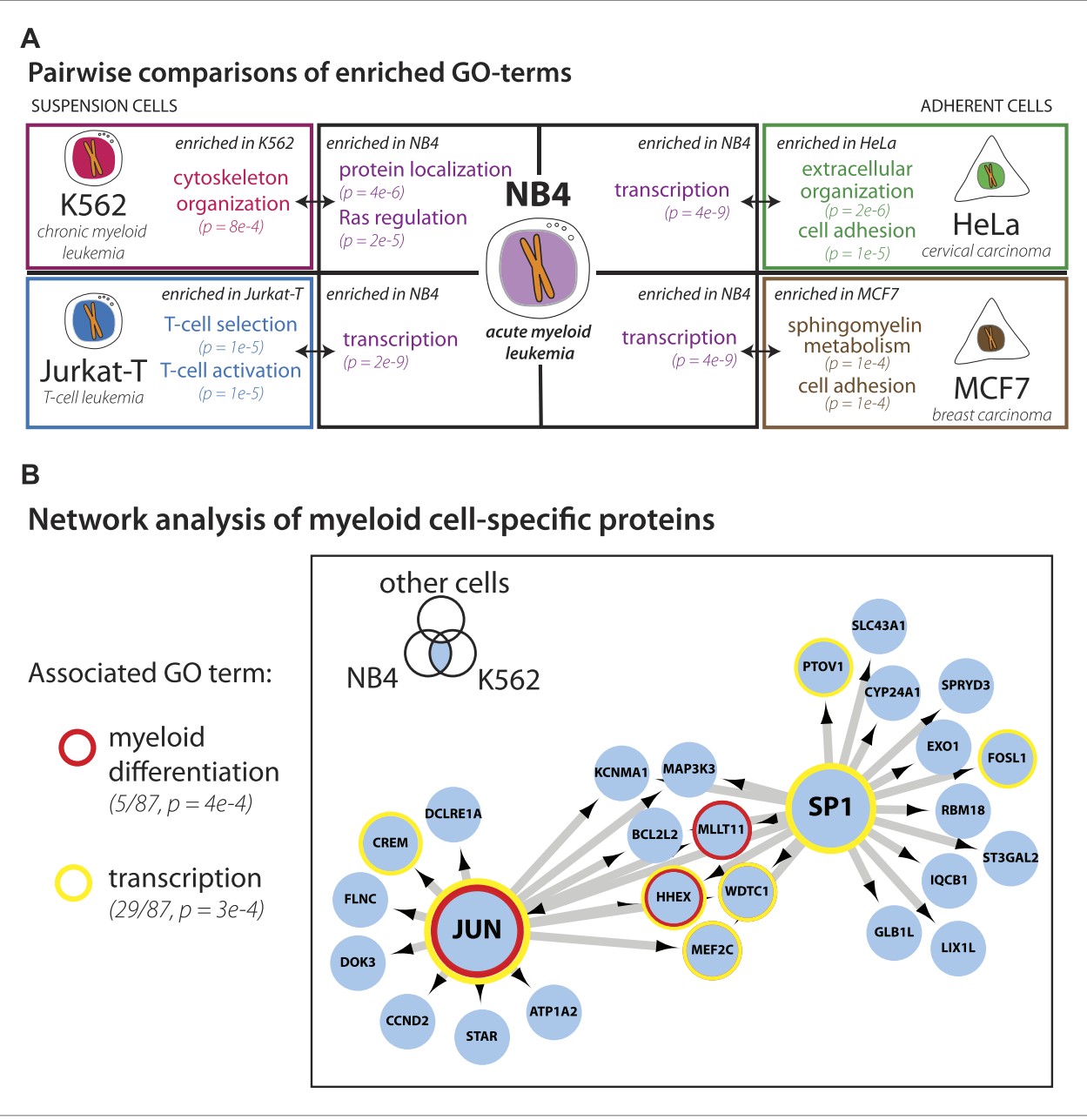

**Figure 4**. Identification of myeloid-specific factors in the NB4 proteome. (**A**) Pairwise comparisons between the NB4 proteome (this study, acute myeloid leukemia) and K562 (chronic myeloid leukemia), Jurkat-T (T-cell leukemia), HeLa (cervical carcinoma), and MCF7 (breast carcinoma) proteomes published by ***Geiger et al. (2012)***. Enriched gene ontology terms and the enrichment p-values are shown for each pairwise comparison. The observed cell-line specific gene ontology enrichments are consistent with the developmental origins of the cell lines (immune vs epithelial), and culturing conditions (suspension vs adherent). The NB4 proteome is highly enriched in transcription factors when compared to cell lines that are not in the myeloid lineage (Jurkat-T, HeLa, and MCF7), implying that there is set of shared transcription factors between NB4 and K562 that may be myeloid-specific. (**B**) A transcriptional regulatory network analysis of proteins identified in myeloid cells (NB4 and K562). Arrows connect transcription factors with their predicted gene substrates (MSigDB). JUN and SP1 appear to be regulatory hubs that can regulate the expression of numerous NB4- and K562-specific genes (***Friedman, 2002***). Together, these data highlight a protein group that may have important transcriptional regulatory activity in myeloid cells. Circles indicate genes that are annotated as being involved in myeloid differentiation (red) or transcription (yellow).

(***Figure 4B***). Further network analysis using the MSigDB transcription factor binding database (***Subramanian et al., 2005***; ***Matys et al., 2006***), revealed evidence for cross-talk between JUN, SP1, and many of the additional genes whose expression was shared between K562 and NB4 cells, but not observed

in either MCF7 or HeLa (*Figure 4B*). Interestingly, immunohistochemical detection of SP1 protein levels by the Human Protein Atlas project (www.proteinatlas.org) showed ubiquitous expression and especially high levels in hematopoetic and placental tissues (*Uhlen et al., 2010*).

In summary, we have identified a specific set of proteins that are preferentially detected in myeloid cells and that may be important in specifying myeloid cell function. We conclude that deep proteome analysis helps to provide a molecular characterization of cell identity by defining sets of genes uniquely expressed in specific cell types.

## Identification of proteins whose abundance varies across the cell cycle

Next, we studied how gene expression in NB4 cells varies across the cell cycle. To do this, we compared the subproteomes of the six, separate elutriated NB4 cell fractions and analyzed how protein abundance varies between the different cell cycle phases. To increase the accuracy of this analysis, we focused on a subset of the total NB4 proteomic data set for which we have protein abundance measurements in all six elutriated fractions and in the asynchronous samples. The 6505 proteins that meet this requirement are supported on average by over 22 distinct peptides per protein, and have a mean sequence coverage of >45%. An arbitrary fold-change cutoff of 2.0 (1.0 in the $\log_2$-transformed axis), was chosen here as the threshold value for cell cycle-regulated abundance change because we observed that this was sufficient to highly enrich for proteins annotated with cell cycle associated GO terms (p<<0.01), as shown in *Figure 5A*. Using these parameters, we identified a group of 358 proteins whose abundance varies across the cell cycle by twofold or more, corresponding to ~5.5% of the high quality, filtered proteomic data set of 6505 proteins (*Supplementary file 3*).

This group of 358 proteins whose abundance is cell cycle regulated was clustered by intensity profiles, which resulted in seven distinct clusters. Scaled protein expression profiles by cluster are shown both as line graphs (*Figure 5B*) and as a heatmap (*Figure 5D*). Six clusters vary primarily by variations in their maximum expression in different elutriated fractions (clusters 1 through 6). In cluster 7, proteins show a marked decrease in abundance during S-phase and peak in the G2&M and G1 fractions (termed 'G2&M + G1' cluster).

Approximately, half of the proteins whose abundance varies significantly across the cell cycle exhibit peak expression in elutriated fractions 5&6, corresponding to late S, G2, and M phases (*Figure 5C*). Proteins whose abundance peaks in S phase are the second most frequent class (27%), followed by proteins peaking in G1 (17%) and proteins that peak in both G2&M and G1 (7%). A large number of human proteins previously reported to be regulated during the cell cycle were identified in this unbiased data set. For example, proteins involved in origin licensing in G1, such as ORC1 (Origin Recognition Complex 1) and DNA replication factor CDT1, peak in G1. UNG (Uracil-DNA glycosylase), which is involved in DNA repair, and Cyclin A2 peak in S phase. Aurora kinase B and cyclins B1 and B2, which are proteins involved in mitosis, peak in G2&M (*Murray, 2004*; *Musacchio and Salmon, 2007*; *Dephoure et al., 2008*; *Olsen et al., 2010*).

The clusters above differ primarily in peak expression across the six elutriated fractions, which can also be broadly classed as G1-, S-, or G2&M-enriched. The six clusters were grouped into these three broader classifications (excluding cluster 7, which peaks in both fractions 1 and 6) and analyzed for enrichment of gene ontology terms (*Table 1*). Additionally, we tested whether the cyclical regulation of protein abundance may be explained, at least in part, by changes in transcription factor activity across the cell cycle. The UCSC TF database contains the predicted promoter binding sites for many important and well-studied transcription factors and is incorporated into the DAVID gene ontology enrichment tool. Thus, we also examined whether the promoters of genes encoding these proteins are differentially enriched in transcription factor binding sites.

*Table 1* lists the corresponding enriched gene ontology terms and transcription factor binding sites by category. Each category is enriched in distinct annotations, indicating that different types and different functional classes of proteins are being regulated during separate phases of the cell cycle. In general, the GO terms we observe enriched in each phase are consistent with many of the activities and processes known and expected for that stage of cell cycle progression. For example, in the category with proteins whose abundance peaks in S phase, there is a clear enrichment for GO terms associated with DNA metabolic functions, reflecting DNA replication as the major metabolic event in S phase. Additionally, the promoters of genes encoding proteins that peak in S-phase are enriched with respect to predicted binding sites for E2F (*Table 1*), a transcription factor that is known to play major roles in regulating entry into the cell cycle and the G1–S transition (*Mudryj et al., 1991*). Interestingly, we

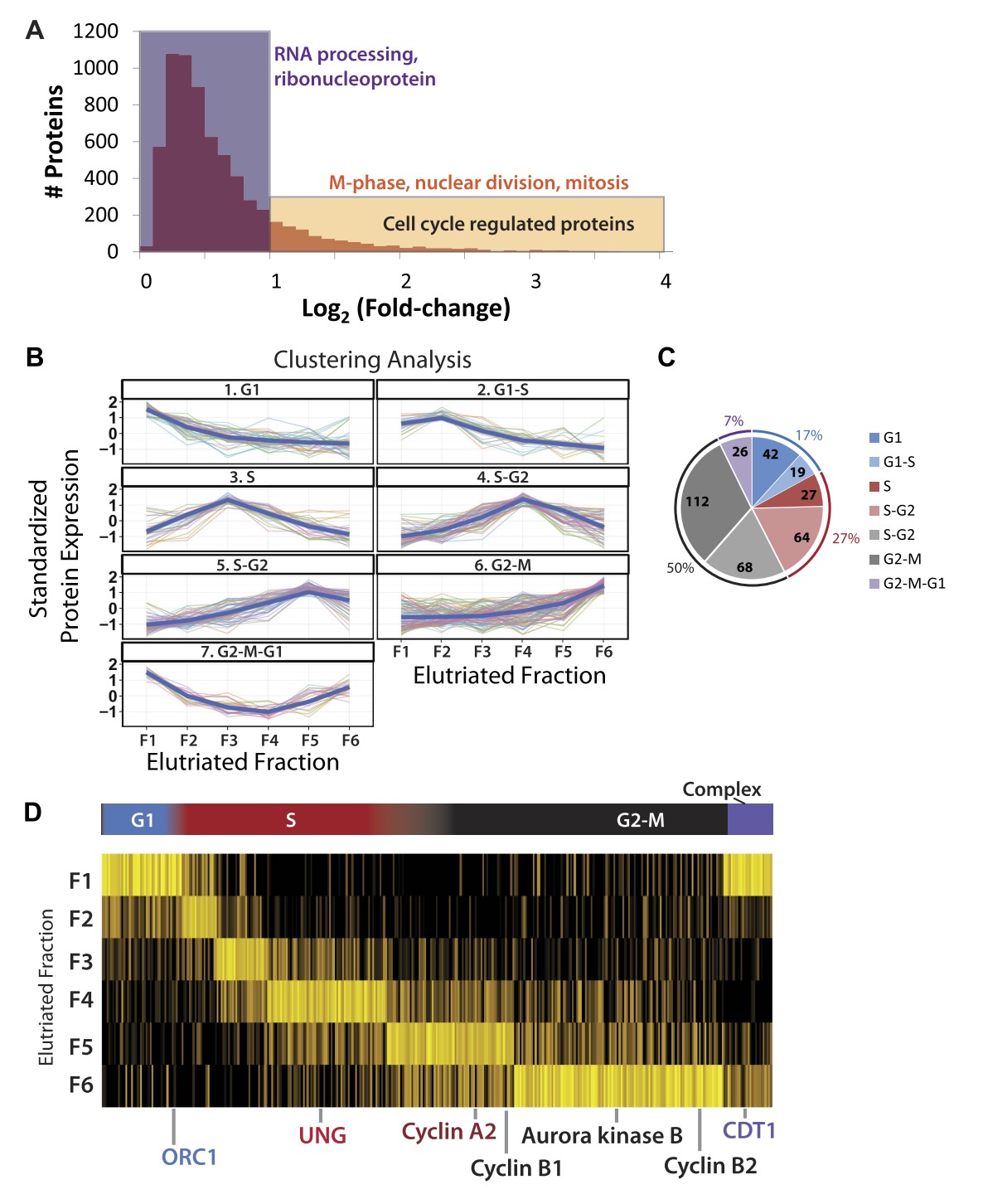

**Figure 5**. Identification of cell cycle-regulated proteins. (**A**) The fold change in label free intensities between any two fractions are shown as a histogram. To identify cell cycle-regulated proteins, an arbitrary fold change cutoff of 2.0 (1.0 in the log2-transformed axis) was set, as indicated by the border between the orange and blue boxes. Highly significant enriched gene ontology terms (p<<0.01) are shown above each group. A twofold change is sufficient to enrich for cell cycle related gene ontology terms, such as M-phase, nuclear division, and mitosis. (**B**) Clustering of the 358 cell cycle-regulated proteins identified in (**A**). Scaled protein intensity profiles were clustered by the phase of maximum expression, with the exception of a small minority of proteins that peaked in multiple phases. Graph titles indicate the phase that is enriched in that fraction. (**C**) The number of cell cycle-regulated proteins split by cluster. Half of the cell cycle-regulated proteins are maximally expressed in the G2&M phase of the cell cycle. (**D**) Scaled intensities are depicted as a heat map. Each vertical line represents a cell cycle-regulated protein, and the shading indicates the intensity (bright yellow being the most intense). Cell cycle regulators established in the literature are highlighted along the bottom of the heat map, and include cyclins A2, B1, B2, and CDT1.

**Table 1.** Enriched functional annotations among the cell cycle varying proteins

| Peak phase | Functional annotation | Proteins | % of total | p value |
|---|---|---|---|---|
| G1 (42 proteins) | transcription cofactor activity | 5 | 12% | 0.003 |
| | transcription factor binding | 5 | 12% | 0.012 |
| S (110 proteins) | phosphoprotein | 82 | 75% | 1.2E-07 |
| | E2F* | 82 | 75% | 0.002 |
| | DNA metabolic process | 10 | 9% | 0.012 |
| | positive regulation of gene expression | 8 | 7% | 0.037 |
| | cell cycle | 11 | 10% | 0.009 |
| G2/M (180 proteins) | M phase | 26 | 14% | 7.5E-20 |
| | cell cycle | 41 | 23% | 3.9E-19 |
| | phosphoprotein | 94 | 52% | 1.8E-06 |
| | NFY* | 93 | 52% | 3.6E-05 |
| Complex (26 proteins) | STAT3* | 20 | 77% | 0.003 |
| | nucleotide-binding | 7 | 27% | 0.015 |

Proteins were partitioned into four categories by peak phase and analyzed for functional annotation enrichment. Functional annotations include gene ontology terms and predicted transcription factor binding sites in the promoter region of the encoding gene. Enriched annotations, their enrichment p values, the number and percentage of proteins with the specified annotation are shown.
*Transcription factor binding sites from the UCSC TFBS database.

identify members of the E2F family (E2F6 and E2F8) as proteins whose abundance peaks in S and G2&M phases, which is consistent with recent reports documenting their role in transcriptional inactivation of G1/S genes (*Bertoli et al., 2013*).

In contrast, the category containing proteins whose abundance peaks at G2&M phase is instead highly enriched for GO terms associated with cell division, M-phase, and NF-Y transcription factor binding sites in their promoters. The category containing proteins that peak in G1 and G2&M (cluster 7), meanwhile, is enriched for genes with promoters that have STAT3 transcription factor binding sites; indeed, 20 out of the 26 genes encoding these proteins (77%), have predicted STAT3 binding sites.

Notably, the GO term 'cell cycle' is only enriched in the S and G2&M clusters. Based on our current data (*Figure 5C*), this may reflect the fact that more cell cycle regulated proteins have peak abundance in S, G2, and M phase than in G1 phase. However, it may also illustrate a feature of the gene ontology annotation system used, linked with the preponderance of previous 'cell cycle' research that has concentrated specifically on analyzing either the entry of cells into mitosis or on studying events during chromosome segregation and mitotic progression and exit. However, our data demonstrate multiple proteins whose abundance is also 'cell cycle regulated' at other stages during interphase, outside of mitosis, which suggests that they should also be annotated with the GO term 'cell cycle'.

## Analysis of protein isoforms whose abundance varies across the cell cycle

In most cases, the protein groups identified by MS analysis correspond to groups of protein isoforms that usually originate from the same open reading frame. However, protein isoforms, even when encoded by the same gene, can have distinct biological properties and can differ in their subcellular localization patterns, interaction partners, and biological functions (*Trinkle-Mulcahy et al., 2006*; *Ahmad et al., 2012*; *Kirkwood et al., 2013*). The high sequence coverage achieved in this NB4 data set improved our ability to discriminate between separate protein isoforms. The peptide data map to 33,575 separates protein isoforms in the Uniprot Human Reference Proteome. Quantitative peptide data, such as either intensities or spectral counts, are normally aggregated into protein groups by sequence similarity and shared peptide evidence. In this study, we have pooled quantitative data by protein isoform, which facilitates the analysis of isoform-specific cell cycle behavior (*Figure 6*). Given that current proteomics workflows achieve ~30 to 40% mean sequence coverage at best, a comprehensive analysis

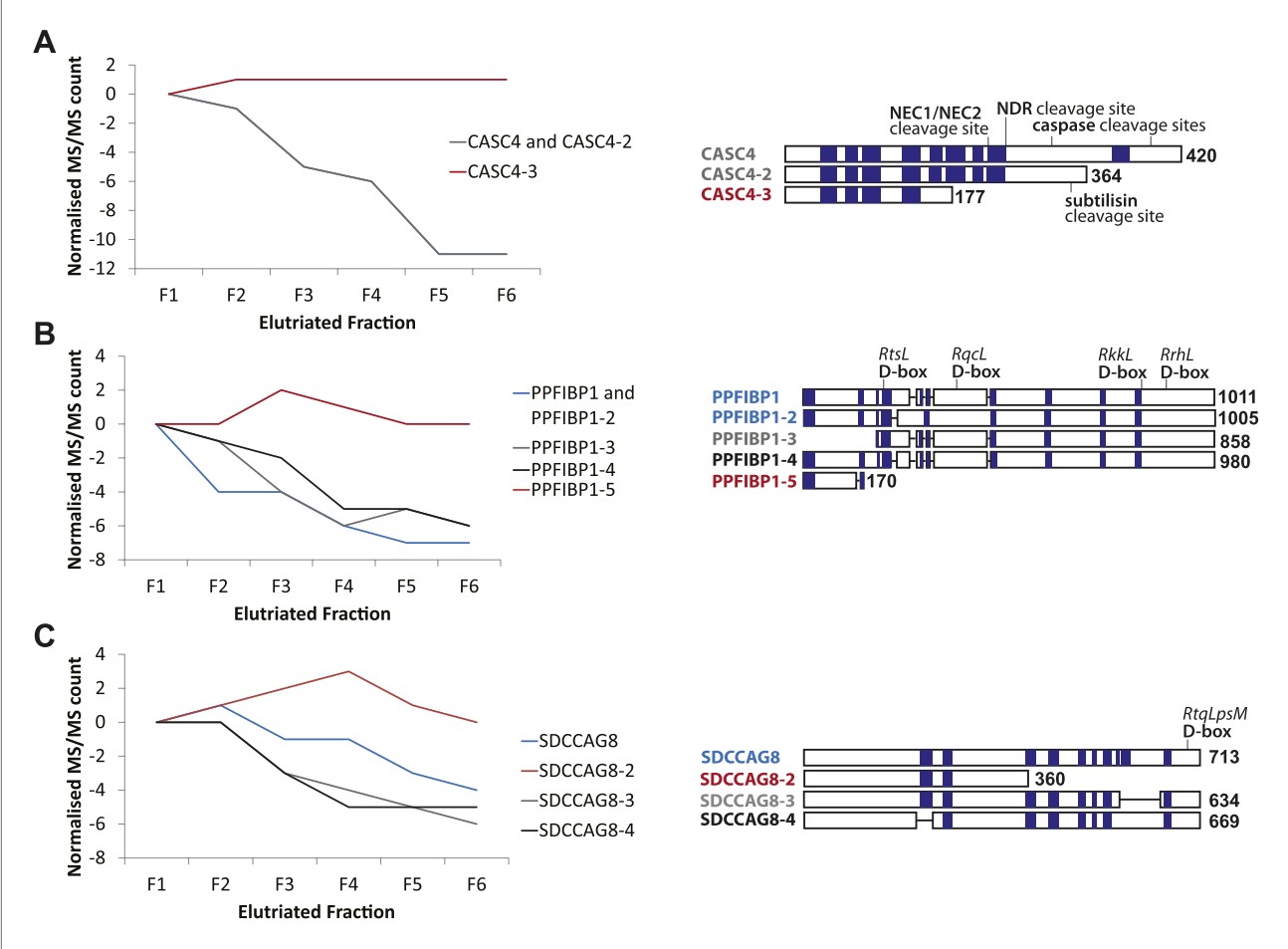

**Figure 6**. CASC4, PPFIBP1, and SDCCAG8 are examples of ORFs that encode multiple splice isoforms that behave differently across the cell cycle. Protein spectral count profiles across the six elutriated fractions for three open reading frames (ORFs) showing protein-level isoform-specific cell cycle variation: CASC4 (**A**), PPFIBP1 (**B**), and SDCCAG8 (**C**). Isoform sequences are shown schematically above each graph. Sequence regions for which direct peptide evidence was detected are shaded in blue, and sequence motifs known to be important in post-translational regulation are indicated.

of isoform-specific variation remains challenging. However, the three examples selected below illustrate the importance of examining isoform-specific protein variation across the cell cycle.

Peptide-level MS data were re-analyzed in the context of predicted and known splice isoforms in the UniProt database (***Kirkwood et al., 2013***). To enhance data quality, we only considered ORFs that have at least two peptides with unique amino acid sequences per protein isoform that were quantified in all six elutriated fractions. These were screened for isoform-specific abundance variation across the cell cycle. Examples of cell cycle-regulated isoforms include proteins encoded by the genes CASC4, PPFIBP1, and SDCCAG8. Line graphs show the spectral counts across the six elutriated fractions (***Figure 6***). Peptide evidences and isoform-specific sequence motifs that target proteins for cleavage and/or degradation, which can affect the activity and abundance of one isoform differentially from the remaining isoforms, are depicted schematically in ***Figure 6***.

We observed differential behavior for CASC4 isoforms, with the abundance of CASC4-1 and CASC4-2 peptides substantially decreasing from G1 to G2&M, whereas CASC4-3 remains invariant (***Figure 6A***). An important point is that simply aggregating the peptide intensities for all of these CASC4 isoforms produces a relatively constant expression pattern across the elutriated fractions. Therefore, conventional bottom-up MS analysis ignoring isoform variation would indicate that the CASC4 ORF generates a polypeptide that is not cell cycle regulated. In contrast, examining the data set with isoform resolution showed isoform-specific cell cycle regulation of separate polypeptides encoded by the CASC4 gene.

PPFIBP1 was initially identified as a cell cycle-regulated protein using the isoform-naive methods described above. However, closer analysis shows that the PPFIBP1 gene encodes multiple isoforms that behave differentially across the cell cycle fractions. Thus, isoforms 1 through 4 vary in abundance across the cell cycle, whereas the abundance of isoform 5 remains relatively constant (*Figure 6B*). Interestingly, several D-box motifs associated with targeted protein degradation are only found in isoforms 1–4, suggesting a mechanism for the isoform-specific cell cycle regulation observed.

The SDCCAG8 gene encodes multiple protein isoforms: three that decrease from G1 to G2&M (isoforms 1, 3, and 4) and isoform 2 that instead peaks in S-phase (*Figure 6C*). As shown with CASC4 above, aggregated peptide data for all isoforms of SDCCAG8 would indicate constant expression across the elutriated fractions, if interpreted as having all of the peptides belonging to a single protein species. However, as a result of the high sequence coverage obtained, it is apparent that the peptides from the SDCCAG8 gene belong to separate isoforms that are differentially regulated across the cell cycle. These three examples underline the value of high peptide sequence coverage that allows isoform-level resolution. The data also indicate that current analyzes likely underestimate the total number of cell cycle-regulated polypeptides, because with current methods many peptides are not detected and thus many isoforms still cannot be reliably discriminated.

## Variation in protein phosphorylation across the cell cycle

Cell cycle progression is controlled not only by changes in protein abundance, but also by other protein properties, including post-translational modifications (PTMs). One of the most important and well-characterized classes of PTM is reversible phosphorylation, which modulates the activity of numerous cell cycle regulatory proteins (*Dephoure et al., 2008*; *Olsen et al., 2010*). In this data set a total of 2761 phosphopeptides were identified, of which 28% were detected with multiple phosphorylated residues in the same peptide (*Supplementary file 4*). Most of the phosphorylation sites identified were on Ser (64%), with the remaining sites evenly split between Thr (17%) and Tyr (16%) (*Figure 7A*). Many of the pTyr sites were found in abundant cytoskeletal proteins, such as actin, myosin, and titin (*Supplementary file 4*). Among the phosphorylation sites that were independently identified by MS/MS in all six elutriated fractions, 89 phosphorylated peptides varied in abundance by more than twofold (~3% of phosphorylation sites detected, *Supplementary file 5*). These sites, which mapped to 79 different proteins, were considered to be cell cycle regulated. Interestingly, only four of these cell cycle-regulated phosphosites mapped to proteins whose abundance varied by more than twofold across the cell cycle (*Figure 7B*). As shown in *Figure 7C*, most of the cell cycle-regulated phosphosites were on Ser (87%). The only cell cycle-regulated phosphotyrosine identified was mapped to Tyr15 of CDK2 and/or CDK1 (both these proteins upon digestion with Lys-C or trypsin, produce the same Tyr15-containing peptide). It has been previously shown that the activities of CDK2 and CDK1 are modulated by differential phosphorylation of Tyr15 at different stages of the cell cycle (*Gu et al., 1992*).

We further identified cell cycle-regulated phosphorylated peptides that peak in abundance at different stages of the cell cycle (*Figure 7D,E*). Four examples of proteins whose phosphorylation status varies across the cell cycle are shown in *Figure 7D*. Several cell cycle-regulated phosphorylation sites were identified in histones (*Figure 7D*, right), which are known to be increasingly phosphorylated from G1 to G2&M (*Sarg et al., 2006*). In the case of TOP2A, which is a protein involved in modulating the topological state of DNA, levels of the peptide phosphorylated at Ser1377 peak in G1/S, whereas an increase in total TOP2A protein abundance is observed from G1 to G2&M. These data suggest two modes of TOP2A activity across the cell cycle: one that is modulated by phosphorylation at Ser1377 during G1/S and a second that may require more copies of TOP2A in the later stages of the cell cycle. In the case of UNG, levels of both the phosphopeptide and the total protein abundance vary in a similar manner across the cell cycle, suggesting that the phosphorylation stoichiometry at this site (Ser23) is relatively constant. In contrast, while the total abundance of TP53 protein is relatively constant, the phosphorylation level of Ser315 on TP53 varies significantly across the cell cycle.

## Transcriptional regulation in NB4 cells across the cell cycle

To investigate the role of transcriptional regulation in the abundance changes observed at the protein level, we undertook a parallel large-scale RNA-Seq analysis of the NB4 cell transcriptome. This was performed both using asynchronous NB4 cells and cells enriched at different cell cycle stages by elutriation. The RNA-Seq analysis was performed both in biological and technical duplicate.

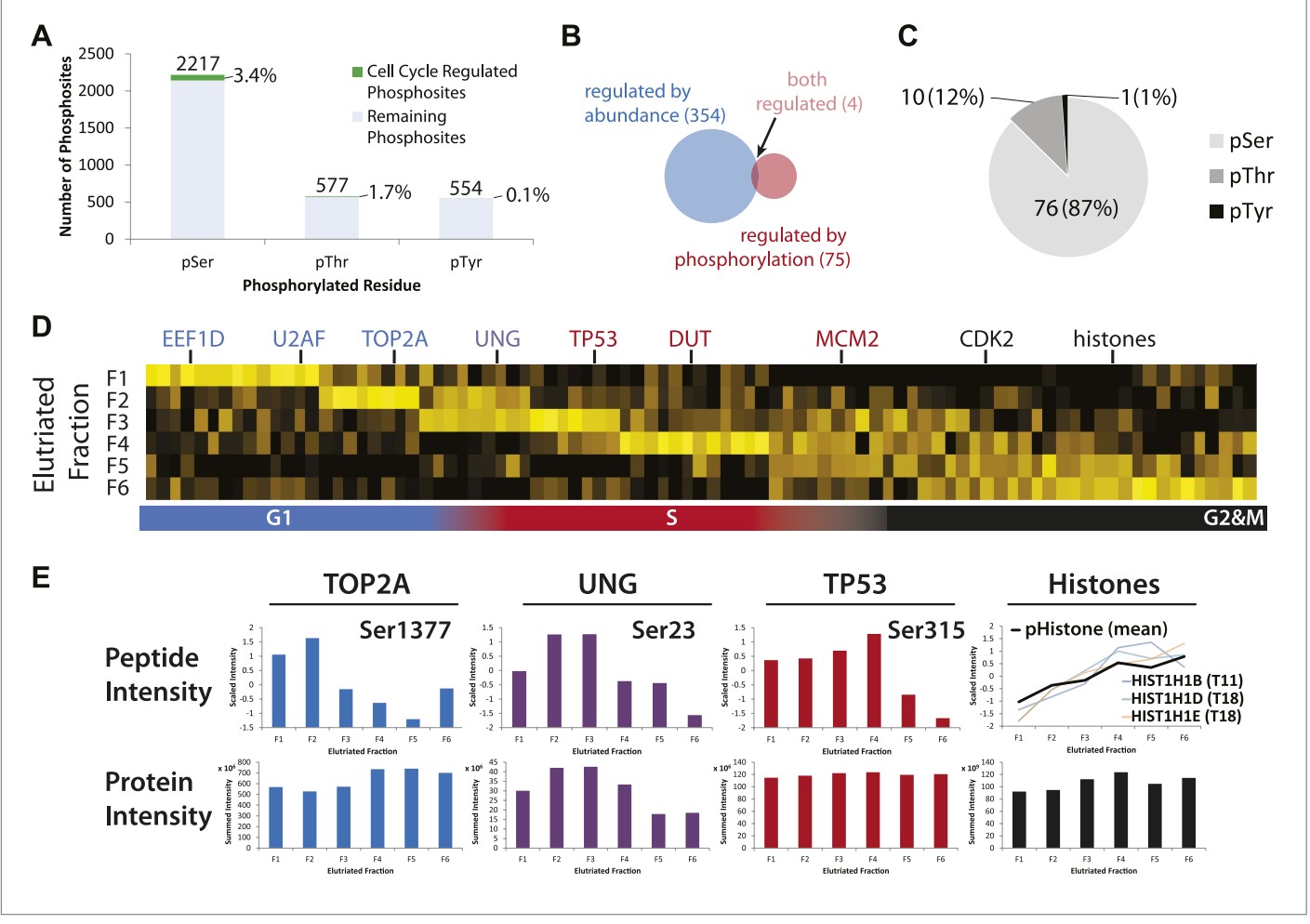

**Figure 7**. Identification of cell cycle-regulated phosphopeptides. (**A**) A total of 2761 phosphorylation sites were identified without phosphopeptide enrichment, which are shown split by residue (Ser, Thr, and Tyr). Cell cycle regulated phosphosites are shown in green. The numbers on top of each bar indicate the total number of pSer, pThr, and pTyr residues detected, respectively. The proportions of cell cycle-regulated pSer, pThr, and pTyr, relative to the total pSer, pThr, and pTyr sites detected respectively, are shown as percentages. (**B**) Overlap between proteins whose abundances are cell cycle regulated and proteins whose phosphorylation is cell cycle regulated. (**C**) A breakdown of cell cycle-regulated phosphosites by residue. The number and the percentage of phosphosites relative to the total number of cell cycle regulated phosphosites are shown. (**D**) Scaled phosphopeptide intensity profiles plotted as a heatmap. Representative cell cycle-regulated phosphorylations that are established in the literature are shown along the top of the heatmap. (**E**) Scaled phosphopeptide and summed protein intensity profiles for four cell cycle regulated phosphorylated proteins (TOP2A, UNG, TP53, and histones). The peptide intensity graphs are annotated with the mapped phosphorylation site. For histones, several phosphopeptide profiles (light purple, light blue, and light orange) and the average (black) are shown on the same graph. The total histone intensity is calculated as the sum of all histones identified.

The combined transcriptomics data set detected mRNA expression from 12,078 genes in NB4 cells (**Supplementary file 6**). This data set was filtered to identify a subset of 9667 genes whose mRNA expression was quantified in each of the three elutriated samples (i.e., combined elutriated fractions F1+F2, F3+F4 and F5+F6). Pairwise comparisons between both the technical and biological replicates show high correlation (r >0.90), (**Figure 8—figure supplement 1**), albeit with higher variance than for replicate proteomics data sets (**Figure 3—figure supplement 1**). Nonetheless, given the overall high degree of reproducibility observed in the biological and technical replicate proteomics and transcriptomics data, we conclude that variance observed comparing protein and mRNA levels (as will be discussed below), predominantly represents biologically significant differences in protein and mRNA expression and cannot be simply explained by variability in either sample preparation and/or measurement error.

Next, we compared directly the expression levels of cognate protein and mRNAs from the same genes. To do this, our detailed proteome (8510 proteins) and transcriptome (mRNA from 9667 genes)

data sets from the asynchronous NB4 cell cultures were merged by mapping the respective protein and gene identifiers to Ensembl Gene IDs. Quantitative protein abundances (iBAQ-scaled, see 'Materials and methods') from expression variants and isoforms that originate from the same gene were aggregated by summation. Histone transcripts were removed from the analysis, due to the known under-representation of poly(A)- mRNAs in our data sets.

In total, we could directly compare the abundances of cognate protein and mRNA for 6170 genes (*Figure 8A*). Qualitatively, this shows that overall, as expected, the levels of protein and mRNA from the same gene are clearly positively correlated. However, the moderate value of the Spearman rank correlation coefficient (0.63), indicates that this positive correlation is not strong enough to consider the level of mRNA alone as a reliable predictor of protein levels for many specific genes. The Spearman rank correlation coefficient of 0.63 measured here is within the upper quartile of the range of previously reported values for correlations between protein and mRNA levels in other mammalian cell types (*Tian et al., 2004*; *Maier et al., 2009*; *Lundberg et al., 2010*; *Nagaraj et al., 2011*; *Schwanhausser et al., 2011*; *Vogel and Marcotte, 2012*).

We also compared the correlation between the expression levels of protein and mRNA from the same genes for the separate data sets derived from NB4 cell populations enriched by elutriation for G1, S, or G2&M phases (*Figure 8B*). This shows that there is little or no difference in the degree of correlation between protein and mRNA, with the same Spearman rank correlation coefficient (r = 0.65) at each of the G1, S, and G2&M phases. These correlation values are similar to what we measured in the asynchronous NB4 cell data set. We conclude that there is little or no global change in the overall relationship between bulk mRNA and protein levels in NB4 cells at different stages of interphase. Furthermore, based on our measurements and correlations of protein and mRNA levels across the elutriated fractions, the data are consistent with there being a large contribution from post-transcriptional mechanisms to controlling gene expression in NB4 cells.

## Correlation of protein and mRNA expression patterns is cell cycle dependent for cell cycle-regulated proteins

Having established that absolute abundances of protein and mRNA are moderately correlated, and that this correlation is independent of cell cycle phase, we next compared specifically the protein and mRNA expression of the 358 proteins whose abundances are cell cycle regulated, as identified in our proteomic data set. First, we examined whether the correlation of absolute protein and mRNA abundances in asynchronous cells is different for genes encoding proteins whose abundance is cell cycle regulated. As *Figure 8C* shows, the protein and mRNA correlation for these genes (r = 0.47) is weaker than the overall correlation for all expressed genes (0.63–0.65). These data suggest that post-transcriptional mechanisms may contribute to a larger extent in cell cycle-regulated gene expression, as compared with bulk gene expression.

We next separately compared the protein and mRNA levels in asynchronous NB4 cells for the four previously determined protein clusters (see *Figure 5*), which differ primarily by the phase in which peak expression occurs (i.e., G1, S, G2&M or G2&M+G1). As *Figure 8D* shows, correlation of protein and mRNA abundances is moderate (r = 0.57) for cell cycle-regulated proteins that peak in either G1 or S. In contrast, protein and mRNA abundances are more poorly correlated for proteins that peak either in G2&M (r = 0.31) or in G1 and G2&M, that is the G2&M+G1 cluster (r = 0.20). This result is not limited to abundances measured in asynchronous cells, as a similar trend is observed when protein and mRNA abundances are compared in elutriated cells at specific phases of the cell cycle, as shown in *Figure 8—figure supplement 2*. For example, the correlation coefficient is 0.42 for G2&M-peaking proteins in G2&M-phase elutriated cells, as compared with values of 0.67 and 0.73 for G1-peaking proteins in G1-phase elutriated cells and S-peaking proteins in S-phase elutriated cells, respectively.

These data show that for cell cycle-regulated proteins, the extent to which protein and mRNA abundance correlates is dependent on when during the cell cycle maximum expression occurs. Thus, cell cycle-regulated proteins that peak in G2&M and G2&M+G1 are more likely to have poor protein/mRNA abundance correlation than cell cycle-regulated proteins whose abundance peaks in other phases of the cell cycle. These results indicate that mRNA levels are particularly poor predictors of protein abundance in the case of cell cycle-regulated proteins whose expression peaks in either G2&M or G2&M+G1.

Next, we examined how well the protein and mRNA expression profiles were correlated across the elutriated fractions. Mechanistically, these patterns result from the combined effects of differential

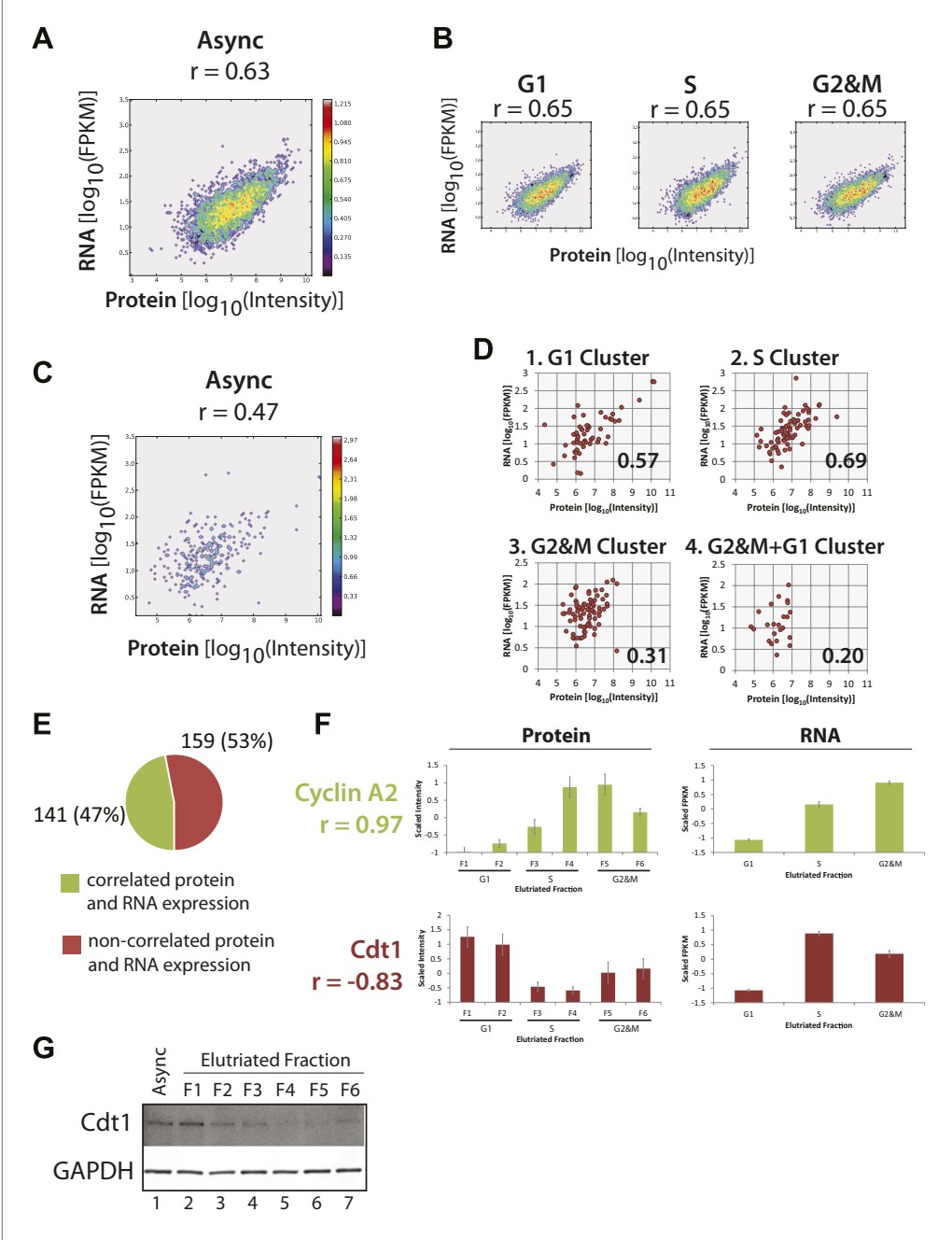

**Figure 8**. Correlation of protein and RNA levels across the cell cycle. Log-transformed, iBAQ-scaled protein intensities and log-transformed FPKM values (RNA) from asynchronous cells (**A**) and cell cycle fractions (**B**). RNASeq data are expressed as Fragments Per Kilobase of exon per million fragments Mapped (FPKM), which is a proxy for RNA copy number. Histone genes were removed from the analysis due to the absence of poly(A)+ tails in histone-encoding messages. Each graph is annotated with the calculated Spearmen correlation coefficients (r). (**C**) Correlation of the protein and mRNA abundances in asynchronous cells of the 358 proteins whose abundances are cell cycle regulated (r = 0.45). (**D**) The same data shown in (**C**), but split by protein clusters as described in *Figure 5*. (**E**) Correlation of the expression profiles of the 358 cell cycle regulated proteins and their associated transcripts. Genes were classified into two groups based on protein and RNA expression correlation (Pearson's correlation coefficient greater than or equal to 0.5). (**F**) Cyclin A2 and Cdt1 are examples highlighted from the groups in (**E**). Protein and RNA abundance standard errors were calculated from the variance in scaled

*Figure 8. Continued on next page*

*Figure 8. Continued*

peptide intensities and biological replicates, respectively. (**G**) Immunoblot analysis of Cdt1 and GAPDH protein expression across asynchronous and elutriated NB4 cells.

The following figure supplements are available for figure 8:

**Figure supplement 1**. Analysis of technical and biological variance among duplicates reveals highly reproducible RNA quantitation.

**Figure supplement 2**. Correlation of protein and RNA abundances of cell cycle-regulated proteins.

---

transcriptional, translational, and post-translational regulation of gene expression across the cell cycle. Highly concerted protein and mRNA expression levels will typically result from synergistic transcriptional and post-transcriptional regulation. Several genes have been previously found to show highly correlated protein and mRNA expression across the cell cycle, including A-type cyclins. Consistent with previous work in other systems (***Pines and Hunter, 1990***), we find that Cyclin A2 protein and mRNA expression is highly correlated in NB4 cells (***Figure 8F***, top). Detailed analyzes have documented regulatory mechanisms that underlie this correlation: namely, cell cycle-regulated transcription of the cyclin A gene, with cyclin A mRNA synthesis being highest in S and G2 (***Henglein et al., 1994***), and destabilization of cyclin A mRNA and protein during early mitosis and G1 (***Pines and Hunter, 1990***; ***Glotzer et al., 1991***; ***Henglein et al., 1994***; ***Dawson et al., 1995***; ***Sudakin et al., 1995***). These synergistic regulatory mechanisms result in differential expression of the cyclin A gene across the cell cycle. Nearly half of the cell cycle-regulated proteins identified in our proteomic data set showed moderate to high correlation (r > 0.5) between protein and mRNA expression patterns (***Figure 8E***). Expression of these genes, like cyclin A, may also be controlled by concerted regulatory mechanisms.

In contrast, over half of the proteins whose abundance is shown here to be regulated across the cell cycle have low correlation (r < 0.5) between protein and mRNA expression patterns. Cdt1 gene expression is even anti-correlated in terms of protein and RNA levels (***Figure 8F***, bottom). Cdt1 mRNA abundance peaks in S-phase, where Cdt1 protein expression is lowest. This surprising inverse relationship between protein and mRNA levels is not likely to be due to low data quality for the Cdt1 gene. A high correlation (>0.75) is observed among the eight, independently identified peptides used to quantify Cdt1 protein expression and error bars indicate the relatively low standard error in scaled intensities from the eight supporting peptides. Furthermore, the Cdt1 protein expression pattern across the cell cycle is confirmed by immunoblot analyzes of elutriated NB4 cells (***Figure 8G***), and is consistent with what is known in other cell types (***Wohlschlegel et al., 2000***). Similarly, the Cdt1 mRNA expression pattern is reproducibly detected, as indicated by the error bars for mRNA quantitation and is consistent with recent reports showing that Cdt1 mRNA levels are high in S-phase due to positive regulation by geminin (***Ballabeni et al., 2013***).

In summary, this study of gene expression in NB4 cells indicates that, even though expression profiles for many genes are positively correlated at the transcript and protein level, for a surprisingly large fraction of human genes mRNA abundance alone is not a reliable predictor of the corresponding abundance of the protein encoded by that mRNA.

## Genes whose protein and mRNA abundances are cell cycle regulated are coordinately expressed across the cell cycle

From the data described above, we determined that protein and mRNA expression are moderately correlated with respect to absolute abundance, that the correlation for bulk gene expression is primarily cell cycle independent, but that over half of the proteins identified as cell cycle regulated have discordant mRNA expression patterns. Out of the 358 proteins whose abundances are cell cycle regulated, 31 of the cognate mRNAs also vary across the elutriated fractions by more than 1.5-fold. Comparison of protein and mRNA abundance profiles across the elutriated fractions for these genes reveals highly coordinated expression (***Figure 9A,B***), as measured by the high Pearson correlation coefficient calculated for the mean protein and RNA profiles (r = 0.93). For each of these 31 genes, protein, and RNA abundances in asynchronous (***Figure 9C***) and elutriated G1, S, and G2&M cells are also positively correlated (r = 0.76, 0.68, 0.79 and 0.84, respectively). These data show that specific subsets of genes can be highly coordinately expressed at both the protein and mRNA level.

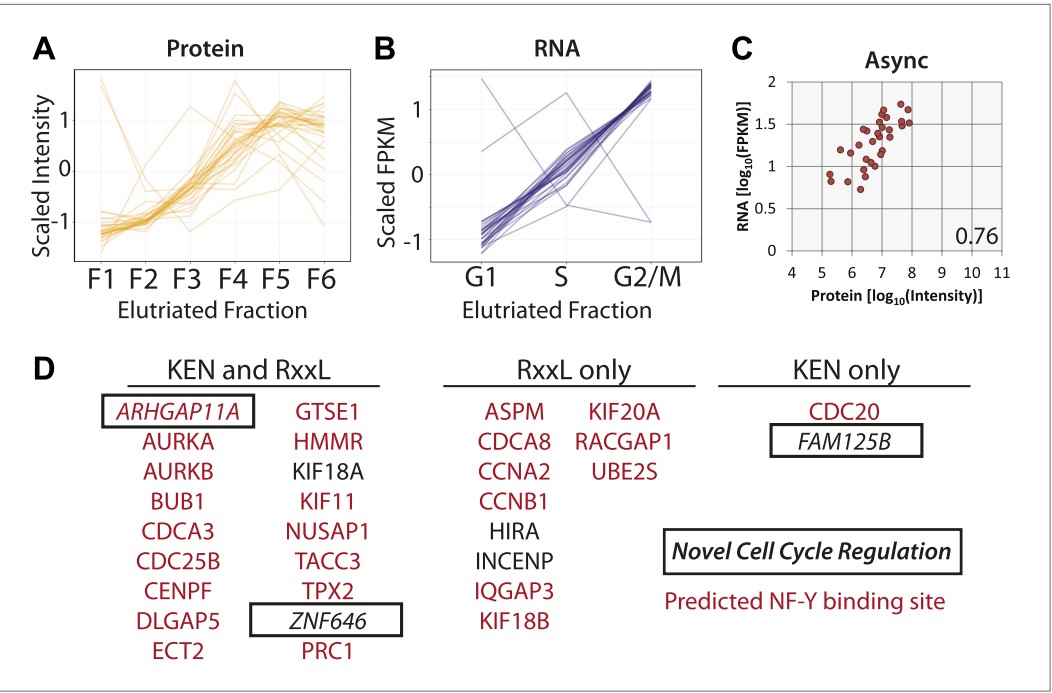

**Figure 9**. Protein and RNA levels are correlated for the specific subset of cell cycle-regulated proteins whose cognate mRNA change by 1.5-fold. Of the 358 proteins whose abundances are cell cycle regulated, the cognate mRNA of 31 proteins also changes across the elutriated fractions by more than 1.5-fold. Scaled protein (**A**) and mRNA expression profiles (**B**) are shown as line graphs for these 31 genes, respectively. (**C**) Comparison of protein and mRNA abundances in asynchronous cells reveals a Spearman correlation of 0.76. (**D**) 27 of the 31 genes have predicted NF-Y binding sites in their promoters, and all 31 encode proteins containing a KEN or D-box sequence degron. KEN-motifs are especially enriched (>eightfold enrichment, compared to 7% expected by random chance). Three genes have not been previously annotated as being cell cycle regulated: FAM125B, ZNF646, ARHGAP11A.

We note that for genes whose protein and mRNA levels covary across the cell cycle, coordinated synthesis of protein and mRNA is likely matched by coordinated protein degradation, as it is known that many proteins that are required at high levels in G2&M phase are targeted for degradation by the Anaphase Promoting Complex/Cyclosome (APC/C) complex during the late stages of mitosis and G1 (**Amon et al., 1994**). Sequence analysis shows that all of these 31 genes contain at least one sequence motif that is known to target proteins for degradation by the APC/C (KEN or RxxL), and most (18/31) contain both motifs (**Figure 9D**). Most of these genes (28/31) have been previously annotated in the literature as either cell cycle regulated and/or critical for cell cycle progression. Additionally, we observe that 26/31 have predicted NF-Y transcription factor binding sites in their promoters (84%), which is significantly more frequent than random (p=2.1 × 10⁻⁶). The association with NF-Y transcription factor binding sites in the promoter region is less frequent for G2&M-peaking cell cycle-regulated proteins (~50%, as shown in **Table 1**, ~49% if the 31 co-regulated genes are excluded), though this is also higher than random (~20% across all promoters).

## ARHGAP11A is a cell cycle-regulated gene whose protein levels are regulated by the APC/C

Among the 31 genes identified whose protein and mRNA abundances are both coordinately cell cycle regulated are three genes (ARHGAP11A, ZNF646, and FAM125B) that have not been previously annotated as being cell cycle regulated (**Figure 9D**). We chose to characterize ARHGAP11A further, as it was the only gene coding for a protein for which validated antibodies were readily available. ARHGAP11A encodes a protein that is predicted to function as a RhoGAP, and has been recently shown to be important in regulating formation of the cytokinetic furrow (**Zanin et al., 2013**). **Figure 10A** shows the MS and RNA-Seq quantitation for ARHGAP11A, indicating that the protein and mRNA abundances are lowest in the first elutriated fractions (G1) and peak in the last elutriated fractions (G2&M). These

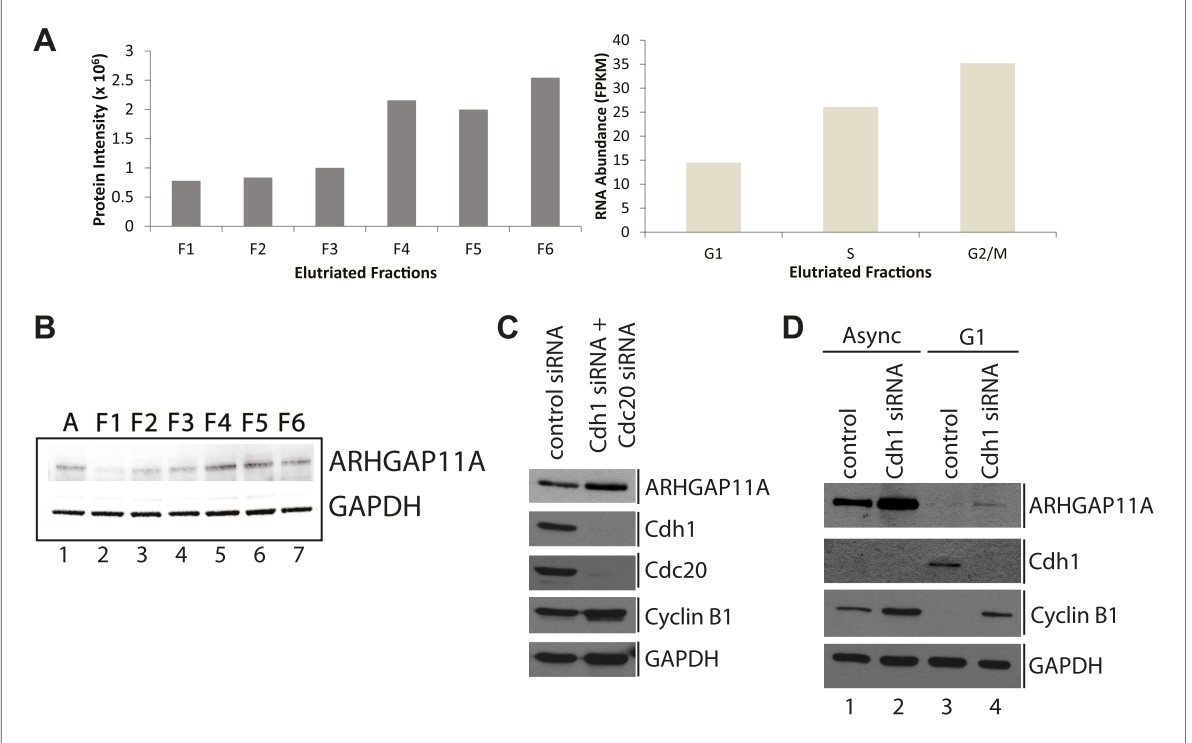

**Figure 10**. Identification of ARHGAP11A as a cell cycle regulated protein and a substrate of the APC/C. (**A**) MS and RNA-Seq quantitation for ARHGAP11A protein (left) and mRNA (right), respectively. (**B**) Immunoblot analysis of ARHGAP11A (HPA antibody) and GAPDH protein expression across asynchronous and elutriated NB4 cells. (**C**) Lysates from U2OS cells treated with either a non-targeting control siRNA (lane 1) or siRNAs targeting Cdh1 and Cdc20 (lane 2) were probed for levels of ARHGAP11A, Cdh1, Cdc20, cyclin B1, and GAPDH by immunoblot. (**D**) Asynchronous or serum-starved RPE-1 cells were treated with either a non-targeting control siRNA or an siRNA against Cdh1. Lysates were then probed with antibodies against ARHGAP11A (Bethyl antibody), Cdh1, cyclin B1, and GAPDH.

The following figure supplements are available for figure 10:

**Figure supplement 1**. Validation of anti-ARHGAP11A antibodies by siRNA-based depletion of ARHGAP11A protein.

cell cycle-dependent changes in ARHGAP11A mRNA abundance are consistent with previous microarray studies performed in HeLa cells (*Whitfield et al., 2002*). The variations in protein levels observed by MS were confirmed independently by immunoblot analysis of elutriated lysates (*Figure 10B*), using a specific antibody that was validated by siRNA depletion (*Figure 10—figure supplement 1*). In contrast, GAPDH does not vary in abundance across the elutriated lysates. Thus, the MS and immunoblot data both show that ARHGAP11A is a cell cycle-regulated protein whose expression peaks in G2&M and is lowest in G1.

Given that the expression of ARHGAP11A is the lowest in G1 and that the protein sequence contains several degrons (one KEN and two D-box motifs), we next tested whether levels of ARHGAP11A protein are regulated by the APC/C. The APC/C is a multimeric protein complex whose activity and specificity is cell cycle regulated by interactions with co-activating factors Cdh1 and Cdc20 (*Visintin et al., 1997*). Intriguingly, Cdc20 was among the 31 genes identified as being regulated both at the protein and mRNA level (*Figure 9*). Disruption of the APC/C complex, for example by siRNA-mediated depletion of Cdh1 and/or Cdc20, is expected to stabilize substrate levels. Thus, we assayed whether ARHGAP11A levels are dependent on APC/C activity by immunoblotting lysates from cells where Cdh1 and/or Cdc20 have been transiently depleted in both U2OS and RPE-1 cell lines using siRNAs.

U2OS cells were treated with specific siRNAs that target either Cdh1 or Cdc20, mRNA for degradation. Transient depletion of both Cdh1 and Cdc20 decreases APC/C activity, as evidenced by a stabilization of cyclin B1 protein (*Figure 10C*), whose levels are known to be regulated by the APC/C (*King et al., 1995*). More importantly, disruption of APC/C activity in U2OS cells increases levels of ARHGAP11A (*Figure 10C*). Similarly, transient depletion of Cdh1 in RPE-1 cells resulted in stabilization of the ARHGAP11A

protein in asynchronous cells (*Figure 10D*). G1-arrest by serum starvation results in very low levels of ARHGAP11A. Transient depletion of Cdh1 results in stabilization of ARHGAP11A, thus showing that ARHGAP11A levels are APC/C$^{Cdh1}$-dependent during G1-phase (*Figure 10D*, lanes 3 and 4). The blots were also probed with anti-cyclin B1 antibodies to confirm disruption of APC/C activity (*Figure 10D*). These data independently show that ARHGAP11A levels are lower in G1-phase than in asynchronous RPE-1 cells (*Figure 10D*, lanes 1 and 3), which is consistent with the cell cycle regulation of ARHGAP11A abundance observed in elutriated NB4 cells. Thus, we conclude that ARHGAP11A is a cell cycle-regulated protein, and that its levels are regulated by targeted degradation mediated by the APC/C.

## Data dissemination through the Encyclopedia of Proteome Dynamics

In addition to uploading the raw MS (http://www.ebi.ac.uk/pride/archive/projects/PXD000678) and RNA-Seq data files (https://www.ebi.ac.uk/ega/datasets/EGAD00001000736) to public repositories (PRIDE and EGA for MS and RNA-Seq data, respectively), we have incorporated the entire analyzed protein and RNA data sets from this study into the Encyclopedia of Proteome Dynamics (http://www.peptracker.com/epd/), a publicly-available, searchable web resource (*Larance et al., 2013*). The EPD aggregates proteomics data from this study on the myeloid cell cycle with our previous large-scale studies on protein complexes, subcellular localization and turnover in HeLa and U2OS cells. The EPD facilitates cross-correlation and analysis of protein properties across numerous, multidimensional proteomics studies. Additionally, for any specific protein, users can quickly retrieve all protein properties measured so far by providing a protein identifier, such as a Uniprot ID.

In this study, we highlight as an example the EPD page for cyclin B1 (*Figure 11*), which displays the protein and RNA quantitation across the separate elutriated fractions from this study, and the protein abundance bins to which cyclin B1 belongs. Users can also retrieve any previously determined properties for this protein, such as putative interaction partners, turnover rate and half-life, subcellular localization and estimated abundances in other cell lines.

## Discussion

In this study, we have performed a deep proteomic analysis combined with RNA-Seq to obtain a global map of gene expression in the human myeloid NB4 cell line. In addition, we have undertaken an unbiased re-evaluation of changes in gene expression across the cell cycle at a system-wide level, using an experimental strategy designed to cause minimum perturbation to the physiology of cell cycle progression in the cells being analyzed. The resulting data set provides one of the most detailed descriptions of human gene expression reported to date, including analysis of the regulated expression of protein isoforms and phosphorylation sites during the cell cycle. Expression of over 10,000 genes was detected at the protein level, with high average sequence coverage (~38%) and expression of over 12,000 genes was detected at the mRNA level. For over 6000 genes, high quality data were obtained at both the protein and mRNA level in each of six subpopulations of cells that had been differentially enriched for cells at distinct cell cycle stages using centrifugal elutriation. We identified 358 proteins whose abundance varied by at least twofold in a cell cycle-dependent manner and these genes were grouped into seven distinct clusters that showed peak expression at different cell cycle stages.

Overall, this data set, augmented by meta-analysis of other recent high-throughput proteomic studies of other human cell lines, provides a new insight into cell identity as characterized by gene expression at the protein level. This highlights specific transcription factors and other proteins with potential roles in myeloid cell differentiation and function. The data also highlight the very complex relationship between the levels of protein and cognate mRNA expression. The protein/mRNA abundance relationship varied quite dramatically for different sets of genes and in many cases the level of mRNA was found not to be a reliable predictor of protein abundance, indicating an important role for post-transcriptional mechanisms in the control of myeloid gene expression. To facilitate data sharing and community access, all of the data from this study have been collated in the Encyclopedia of Proteome Dynamics (http://www.peptracker.com/epd), a searchable online database describing system-wide measurements of protein properties (*Larance et al., 2013*).

## In-depth characterization of a minimally perturbed cell cycle

To facilitate analysis specifically of physiologically relevant variations in gene expression across the cell cycle, we developed an experimental strategy that avoided the use of synchronization procedures to accumulate cells blocked at specific cell cycle stages. In previous studies, this is usually done using

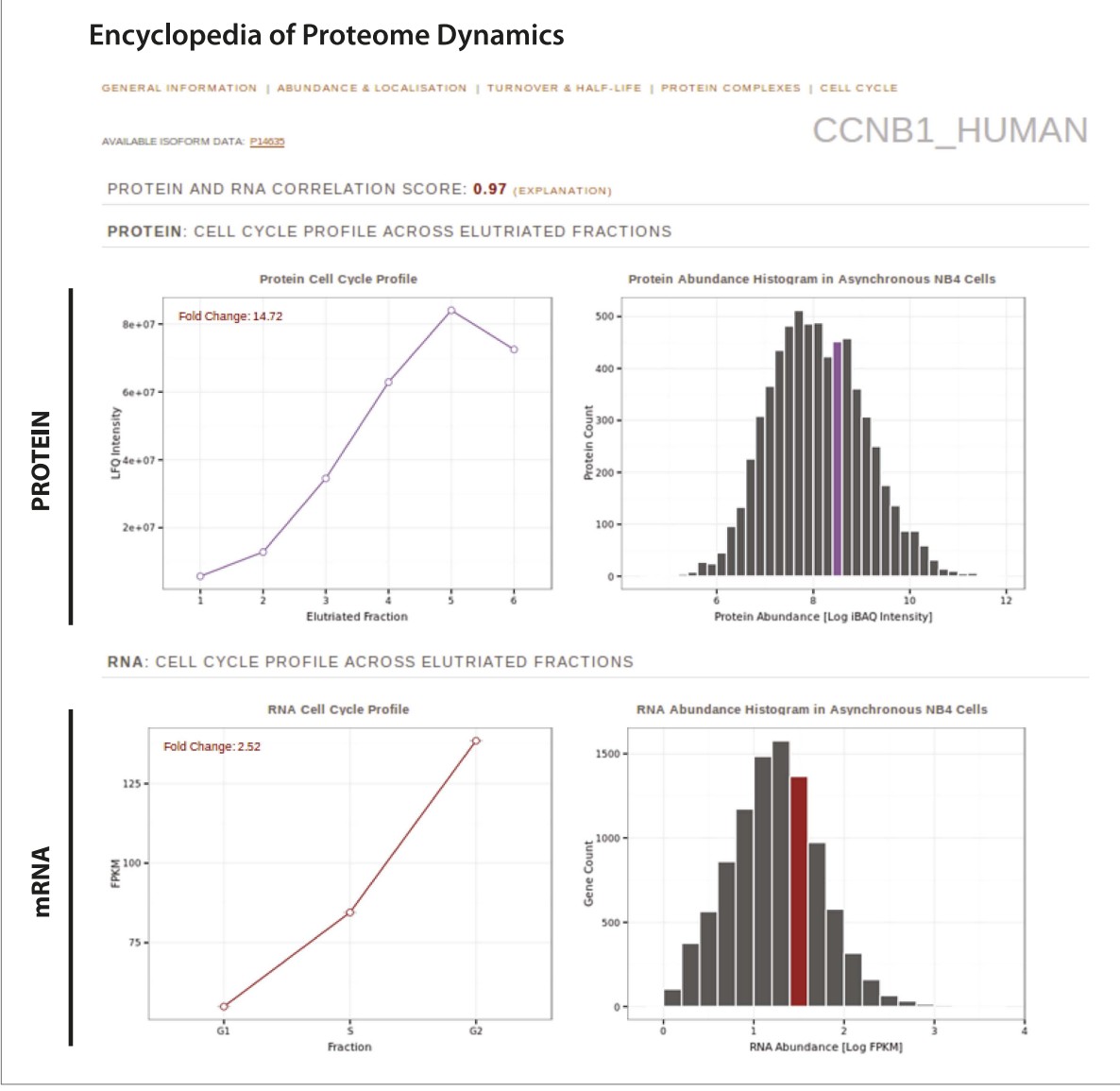

**Figure 11**. The Encyclopedia of Proteome Dynamics, a fully searchable, open-access online repository of proteome data. Quantitative protein and RNA data from this study are available through the Encyclopedia of Proteome Dynamics (EPD). A screenshot of the EPD is shown, which displays protein and mRNA expression profiles across the elutriated fractions, the calculated Pearson correlation coefficient between the protein and mRNA profiles, and protein and mRNA abundances in asynchronous cells for cyclin B1 (CCNB1_HUMAN).

either chemical inhibitors or genetic depletion of essential factors, to either activate checkpoints or otherwise block progression through the cell cycle, thus allowing a large enough population of cells at the same cell cycle stage to be harvested for subsequent biochemical analysis. Although effective, the potential disadvantage of these approaches is that they inevitably cause a major metabolic perturbation and/or stress to achieve the cell cycle block and this in turn may affect gene expression in ways that would not occur during normal cell cycle progression. In addition, to minimize possible effects of specialized media on the normal cell cycle, we also avoided here using cell media with dialyzed serum, as frequently used for metabolic labeling of cells with heavy isotope substituted amino acids. Consequently, we focused on growing cells in normal sera and identified proteins by performing label-free MS analysis and we used centrifugal elutriation as the method to generate sub-populations of cells enriched at different cell cycle stages.

Centrifugal elutriation is a simple yet effective physical method of enriching for cells at different cell cycle phases that is suitable for isolating sufficient quantities of cells for large-scale biochemical analysis

(*Banfalvi, 2008*). We used a variety of criteria here, including FACS analysis and immunoblotting, to detect multiple known cell cycle-regulated and control proteins, thus validating the successful separation of NB4 cells into six fractions that were differentially enriched for cells at different cell cycle stages. Compared with using metabolic inhibition to block cells at different cell cycle stages, elutriation generates minimal stress or disruption of cellular metabolism and physiology. We confirmed that here by showing that the elutriated NB4 cells remained viable and continued to grow and proliferate when returned to culture post elutriation without showing evidence of changes in morphology. Because the separation principle of elutriation is based on physical size, we note that this method can also be used to separate cells of different sizes and thus could also be employed to examine changes in either gene expression or other processes, associated with cell size variation (*Bjorklund et al., 2006*; *Tzur et al., 2009*; *Navarro et al., 2012*).

In combination with centrifugal elutriation, we thus determined both protein and mRNA levels for more than 6000 human genes in each of the six, separate elutriated fractions. Analysis of these data highlighted examples of three major types of protein cell cycle regulation, that is, changes in protein abundance, isoform-specific changes in abundance and changes in protein phosphorylation.

We identified ~5% of genes as encoding proteins whose abundance varies across the cell cycle by at least twofold. These genes formed seven distinct clusters based upon the cell cycle stage where the protein showed maximum expression level. As this study represents a re-evaluation of cell cycle-regulated gene expression and was not influenced by expectations from the literature, it was reassuring that it identified and confirmed so many previously documented cell cycle-regulated factors. In addition, it detected novel cell cycle regulated genes and showed that the current gene ontology annotations are primarily associated with genes in the clusters showing peak expression at either entry or exit from mitosis. However, other clusters show collections of genes that also express proteins whose abundance is regulated at other stages of interphase and we propose that these should also be annotated in future as 'cell cycle regulated' and that this term should not be restricted to genes linked with mitosis.

Our analysis of the genes encoding proteins whose abundance is cell cycle-regulated implicates several transcription factors as being particularly important for the control of protein expression during the cell cycle. For example, in addition to recapitulating the previously described importance of E2F transcription factor activity in S-phase (*Sherr, 1996*), these data also highlight the potentially important roles for the NF-Y and STAT3 transcription factors for gene regulation in G2&M and G1&G2, respectively. The enrichment of NF-Y binding sites we observed in the promoters of genes encoding proteins that peak in the G2&M fraction is consistent with previous reports linking NF-Y transcriptional activity to G2&M phase cell cycle functions (*Hu et al., 2006*).

In comparing our present results with previous data on cell cycle-regulated gene expression in human cells, it is interesting that the rather low number of cell cycle-regulated proteins detected here in elutriated NB4 cells (~5%), contrasts with a much higher estimate of ~40% of the detectable proteome varying by at least twofold in abundance across the cell cycle in HeLa cells (*Olsen et al., 2010*). This proportion was determined using thymidine- and nocodazole-synchronized HeLa cells and is significantly higher than what was reported for HeLa cells synchronized by thymidine alone (~15%) (*Lane et al., 2013*). The difference in the proportion of cell cycle-regulated proteins observed between these studies may be due to technical differences, analysis criteria, growth conditions, the type of quantitation employed and/or the synchronization method. Differences between studies performed in HeLa and NB4 cells may also reflect tissue-specific and/or cell line-specific plasticity in cell cycle regulation in mammalian systems.

## Comparison of protein and mRNA expression levels

The data in this study allowed us to examine in detail the relationship between protein and mRNA abundance, both in terms of global genomic expression and for specific sets of genes and for cells at different stages of cell cycle progression. The results showed that this relationship is remarkably complex and clearly indicate that measurements of mRNA levels alone cannot be relied upon to provide an accurate reflection of protein abundance in many situations. Indeed, as illustrated for the Cdt1 gene, we identify examples where protein and mRNA levels are even inversely correlated. The rather complex relationship we observe between protein and mRNA levels is important for interpreting previous studies that have relied on using either microarray or RNA-Seq data alone as a surrogate for reporting on the regulation of protein expression.

Our data on NB4 cells show that for over 6000 genes protein and mRNA abundances are positively correlated, both in asynchronously growing cells and in cells at G1, S, and G2&M phases, though in each case with a moderate correlation coefficient of ~0.63 to 0.65. However, this correlation is weaker (0.47) for the ~5% of genes encoding proteins whose abundance varies across the cell cycle. Interestingly, this subset of cell cycle-regulated genes shows further heterogeneity in the relationship between protein and mRNA abundance levels when examined more closely to compare the separate clusters of genes encoding proteins whose expression peaks at different stages. Thus, we observe dramatically varying correlations in protein/mRNA abundance levels in different clusters (*Figure 8*). This ranges from the very low value of ~0.2 ('G2&M+G1'-peaking proteins, cluster 4), up to the higher than average value of ~0.69 for proteins with peak abundance in S phase (cluster 2).

## Transcriptional and post-transcriptional regulation

The NB4 cell data show that a subset of the genes that encode proteins whose abundance is cell cycle regulated is concordantly expressed at the protein and mRNA level. This is the case particularly for genes encoding proteins whose abundance peaks in either S or in the G2&M phases of the cell cycle. This suggests that transcriptional regulation mechanisms contribute significantly to modulating protein expression levels in these phases, but less so in G1. A likely explanation for this observation, consistent with the literature on cell cycle regulatory mechanisms, is that post-transcriptional controls, including targeted protein degradation and regulation of translation, operate differentially across the cell cycle and cause temporal imbalances in the relation between transcript levels and proteins. It is well known that mechanisms exist for controlling the targeted degradation of specific proteins, for example, based on the substrate-specific action of E3 ligases targeting proteins for degradation by the proteasome (*King et al., 1996*).

A number of cell cycle-regulated proteins have been shown to be targeted for degradation at specific cell cycle stages, as exemplified by mechanisms such as the degradation of proteins by the APC/C at the end of mitosis and into G1 phase (*King et al., 1996*; *Reed, 2003*; *Pines, 2011*). Indeed, recent high-throughput studies of protein turnover in HeLa and U2OS cells demonstrated that the most rapidly degraded group of proteins were strongly associated with the GO terms 'cell cycle' and 'mitosis' (*Boisvert et al., 2012*; *Larance et al., 2013*). In this study, we identify many known cell cycle-regulated APC/C substrates, including aurora kinases (*Honda et al., 2000*) and securin (*Zur and Brandeis, 2001*). In addition, we validated here a novel cell cycle-regulated protein (ARHGAP11A), whose regulation is mediated by targeted degradation by the APC/C.

In addition to the APC/C, whose activity is restricted to mitosis and G1, there are other complexes that target proteins for degradation in other phases of the cell cycle. For example, the SCF$^{Skp2}$ complex targets proteins for degradation during and at the entry of S-phase. Substrates include Cdt1 (*Wohlschlegel et al., 2000*), which is critical for replication origin licensing and Cep192, whose hydroxyproline-modified form is targeted for degradation by the SCF$^{Skp2}$ complex (*Moser et al., 2013*). We note that the clustering analysis identified a subset of NB4 proteins whose abundances are similarly regulated across the cell cycle as Cdt1 (*Figure 5*, cluster 7). Proteins with Cdt1-like expression patterns may be similarly regulated by SCF$^{Skp2}$; however, additional experiments are required to test these hypotheses.

In contrast with the extensive literature on cell cycle-regulated protein degradation, studies examining translational control across the cell cycle, particularly high-throughput studies, are few in number. However, a recent study using puromycin-analogs and in vitro immunoprecipitation in thymidine-synchronized HeLa cells revealed that a subset of the proteome (339/4984 proteins) is differentially translated across the cell cycle, while the translation of most proteins remains relatively constant (*Aviner et al., 2013*). Interestingly, these authors propose that mRNA translation is particularly important in regulating G2&M-associated proteins. It will be interesting in future to examine the contribution of translational regulation mechanisms to the cell cycle variations in protein abundance measured here in NB4 cells.

## In-depth characterization of a myeloid leukemia proteome

We have shown that the use of a dual protease digestion strategy combined with extensive prefractionation of isolated peptides by strong anion exchange-enabled proteomic characterization of a leukemia cell line to a depth of over 10,000 proteins, identified with on average 15 peptides per protein and with high sequence coverage (mean ~38%). To the best of our knowledge, this data set, together with the accompanying RNA-Seq data, provides the most comprehensive study to date of gene expression in

human myeloid cells. The depth of proteome sequence coverage is comparable with, if not exceeding, the proteome coverage obtained in recent deep proteome studies performed on other human tumor cell lines, including epithelial tumor cell lines such as HeLa and U2OS. The proteomic workflow is not exclusive to either suspension cells or leukemia cell lines and, based on current technology and instrumentation, is applicable in principle to any cell type where ~200 μg of protein can be isolated.

The high protein sequence coverage obtained was shown to be particularly helpful for allowing a more detailed comparison of gene expression and regulation at the level of separate protein isoforms. Importantly, using the information from the high number of independent peptide identifications per protein group, we could show clear examples where genes encoded multiple isoforms, only a subset of which were cell cycle regulated in their expression pattern. This also showed that aggregating all of the peptide information and interpreting it in terms of the behavior of a single hypothetical polypeptide, as typically done in proteomic studies, would lead to an incorrect conclusion that the corresponding gene was not subject to cell cycle regulation because the peptide information for the cell cycle-regulated isoform was diluted by the contribution of the peptides shared with the other, unregulated isoforms encoded by the same gene.

We infer that our current data underestimate the total number of polypeptides whose abundance is cell cycle regulated, not only because we lack combined protein and mRNA data across the full cell cycle for at least 4000 additional genes whose expression was detected in the asynchronous NB4 cell populations, but also because we lack sufficient numbers of peptides to be confident that we are efficiently detecting and quantifying most of the separate protein isoforms that are expressed. Our data therefore highlight the need for further technological development in proteomics methods and instrumentation because even in the deepest analyzes reported to date, as with this present study, still less than 50% of the total protein sequence is identified for the genes we can detect being expressed at the protein level. It is also difficult to detect multiple peptides consistently across every sample in a complex experiment, as is the case here with multiple elutriated cell populations. Nonetheless, we anticipate that continued improvements in instrumentation, combined with improved sample preparation, will provide additional sequencing depth and speed in the future and the present study illustrates how this can be used to produce a more comprehensive mapping of gene expression and regulation during fundamental biological processes.

## Towards a proteomic definition of cell identity

A meta-analysis of the data from human cell lines where the most detailed proteomic information is available revealed a set of proteins whose expression was only detected here in NB4 cells. This NB4-specific protein set was enriched for transcription factors, including proteins that are already known to be important for myeloid cell differentiation, such as PU.1 and C/EBPα/δ (*Orkin and Zon, 2008*). We also identified a core set of proteins that were detected in myeloid-derived cell lines (K562 and NB4), but not in other cell lines, most of which are non-leukemic and from epithelial origin. Many genes encoding these proteins are overexpressed at the mRNA level in normal blood, leukemia, and lymphoma cells compared to other normal and tumor tissues in the Broad Global Cancer Map (*Figure 12*) (*Subramanian et al., 2005*), and are enriched in genes that are overexpressed in leukemia/lymphoblastic tissues in the Novartis GNF tissue expression database (25/87 genes, p=0.00059) (*Su et al., 2002*). The proteomic data here provide direct evidence that many of the overexpressed mRNAs observed are translated into protein and that these proteins are likely overexpressed in cancer cells of the myeloid lineage compared to cancer cells derived from other tissues. For comparison, HeLa-specific genes were similarly analyzed and found to be overexpressed in normal uterine tissue and prostate tumors (*Figure 12*). Given the high variance observed between mRNA and protein abundance, we note that direct experimental evidence of protein overexpression has added benefits to clinical pathology and diagnostics.

## Proteomic approach to gene regulation

Gene regulation that extends to the protein level, including cell cycle-dependent regulation, broadly, encompasses the modulation of any properties of proteins and protein isoforms and not just variations in protein abundance, as we have focused on in this study. A more comprehensive analysis of cell cycle regulation of gene expression should thus in future be extended to analyze also variations in the subcellular localization of the proteome, changes in protein complex formation and protein–protein interactions and a more detailed description of protein isoform expression and post-translational protein modifications. Methods are now emerging that should allow the systematic and quantitative analysis of these varied properties at a system-wide level. It will be important also to repeat such in depth

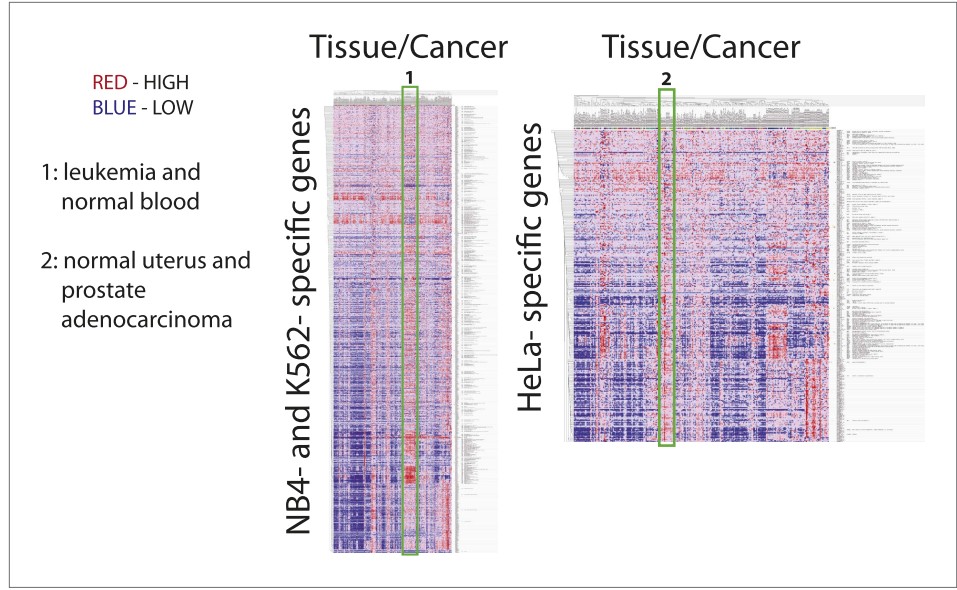

**Figure 12**. Many cell-line specific genes are overexpressed in tumors and normal tissues that are associated with the developmental origin of the cell line. mRNA expression heatmaps from the Broad Global Cancer Map for NB4- and K562- specific genes (left) and HeLa-specific genes (right). Each heatmap has tissue along the horizontal axis and gene along the vertical axis. Vertical red streaks indicate that many genes are similarly overexpressed in a particular tissue. Many NB4- and K562-specific genes are overexpressed in lymphoid, leukemia, and normal hematopoietic tissues, whereas HeLa-specific genes are overexpressed in normal uterine tissues and prostate tumors.

studies on gene expression across a wide range of cell types, particularly also in primary cells, to evaluate what types of mechanisms are used ubiquitously and which are used in conjunction with specific needs of specialized cell types and/or modified by the process of cell transformation and influenced by oncogene expression. It will also be interesting to compare the results of cell cycle-regulated gene expression detected here using centrifugal elutriation to separate cells into distinct populations enriched at different cell cycle stages with comparable analysis of cells, where cell cycle progression is blocked with inhibitors to compare what effect such metabolic perturbations may have on gene expression, separate to the normal control of the cell cycle.

We have collated all of the data from this study into the Encyclopedia of Proteome Dynamics, a searchable online database (http://www.peptracker.com/epd). The data here are combined with other high-throughput studies of protein properties in HeLa and U2OS cells, including the system-wide analysis of protein turnover and protein degradation rates in separate subcellular compartments and the analysis of native, multi-protein complexes separated by size exclusion chromatography (*Ahmad et al., 2012*; *Boisvert et al., 2012*; *Kirkwood et al., 2013*; *Larance et al., 2013*). In common with this present study, all of these data show the importance of moving beyond simple protein identification to a more detailed analysis of complex proteome dynamics, including the analysis of selective regulation of distinct protein isoforms and post-translational modifications for deciphering cellular regulation mechanisms. Furthermore, the combining of highly annotated large data sets in this format not only adds value through sharing of information with the community, it also facilitates a Super-Experiment approach (*Boulon et al., 2010*). Here, the value of each individual data set is enhanced by allowing the detailed cross-comparison and analysis of protein behavior and responses between different cell types and under different growth conditions. We suggest that this provides a useful model that could be extended in future to provide a resource incorporating data generated at a community wide level.

## Materials and methods

### Cell culture and counterflow centrifugal elutriation

The NB4 cell line was established from long-term cultures of acute myeloid leukemia blast cells grown on bone-marrow stromal fibroblasts (Lanotte, 1991). NB4 cells were obtained from the Hay laboratory

(University of Dundee). The cells were cultured at 37°C in the presence of 5% $CO_2$ as a suspension in RPMI-1640 (Life Technologies, United Kingdom) supplemented with 2 mM L-glutamine, 10% vol/vol fetal bovine serum (FBS, Life Technologies), 100 units/ml penicillin and 100 µg/ml streptomycin (100X stock, Life Technologies). Cell cultures were maintained at densities between $1 \times 10^5$ and $1 \times 10^6$ cells/ml and harvested by centrifugation when cultures reached ~$8 \times 10^5$ cells/ml.

Pellets containing ~$5 \times 10^8$ cells were resuspended in 5 ml elutriation buffer (PBS +1% FBS). The resulting cell suspension was passed through a 19G-needle three times to disaggregate cell clumps and then loaded into a Beckman counterflow centrifugal elutriator (Beckman JE-5.0/JE), equipped with a standard elutriation chamber and a Cole–Parmer MasterFlex Model 900-292 peristaltic pump. The centrifuge was operated at 1800 rpm and the flow rate was initially set to 16.68 ml min$^{-1}$. After cells have been loaded into the elutriation chamber, 50 ml fractions were collected at the following flow rates: 21.18, 23.88, 25.68, 27.47, 29.27, and 38.27 ml min$^{-1}$ (i.e., fractions 1 through 7). A 2 ml aliquot containing a minimum of $5 \times 10^5$ cells from each fraction was reserved for flow cytometry. The remaining cells were harvested for protein and RNA extraction.

Cell yields range from >$8 \times 10^7$ cells (Fraction 1, G1) to $2 \times 10^6$ cells (Fraction 6, G2&M), which reflect the typical cell cycle phase distribution found in cultured cell lines. Fraction 6, which yields the lowest cell number, still provides ~1.0 mg of total protein, which is sufficient for in-depth proteomics analysis using current technology (min ~200 µg).

## Single Shot (SS) proteomics sample preparation

For protein extraction, NB4 cells were pelleted, washed twice with cold PBS and then lysed in 0.3–1.0 ml HES lysis buffer (2% SDS, 10 mM HEPES pH 7.4, 1 mM EDTA, 250 mM sucrose, Roche protease inhibitors, Roche PhosStop; UK). Lysates were heated to 95°C for 10 min and homogenized using Qiashredder (Qiagen). 200 µg of the lysate was further processed for LC-MS/MS analysis using a modification of the FASP protocol (*Wisniewski et al., 2009*). Briefly, lysates were loaded onto pre-equilibrated 30 kD-cutoff spin columns (Sartorius UK) and washed twice using denaturing urea buffer (8 M urea, 10 mM Tris, pH 7.4). Proteins were reduced with TCEP (25 mM in denaturing urea buffer), for 15 min at room temperature and alkylated with iodoacetamide (55 mM in denaturing urea buffer), in the dark for 45 min at room temperature. Lysates were then buffer-exchanged into 0.1 M triethylammonium bicarbonate, pH 8.5 (TEAB, Sigma) and digested with trypsin (1:50, Promega UK) overnight at 37°C. Digestion efficiency was checked by SDS-PAGE analysis and protein staining with SimplyBlue SafeStain (Life Technologies). After collecting the first peptide flow-through, the spin column was washed twice with 0.1 M TEAB, then twice with 0.5 M NaCl. The flow-through and washes were combined and desalted using SepPak-C18 SPE cartridges (Waters UK). Peptides were then dried and resuspended in 5% formic acid for LC-MS/MS analysis.

## Strong anion exchange (SAX) proteomics sample preparation

For protein extraction, NB4 cells were pelleted, washed twice with cold PBS and then lysed in 0.3–1.0 ml urea lysis buffer (8 M urea, 100 mM Tris pH 7.4, Roche protease inhibitors, Roche PhosStop). Lysates were vigorously mixed for 30 min at room temperature and homogenized using a Branson Digital Sonifier (30% power, 30 s). Proteins were reduced with TCEP (25 mM in denaturing urea buffer), for 15 min at room temperature and alkylated with iodoacetamide (55 mM in denaturing urea buffer), in the dark for 45 min at room temperature. Lysates were diluted with digest buffer (100 mM Tris pH 8.0 + 1 mM $CaCl_2$) to reach 4 M urea, and then digested with 1:50 Lys-C (Wako Chemicals, Japan) overnight at 37°C. The digest was then split into two fractions. The first was retained as the Lys-C digest, which was shown previously to produce peptides that are complementary to trypsin (*Swaney et al., 2010*). The second was diluted with digest buffer to reach 0.8 M urea and double-digested with trypsin (Promega; 1:50). Digest efficiencies were checked by SDS-PAGE analysis and protein staining. The digests were then desalted using SepPak-C18 SPE cartridges, dried, and resuspended in 50 mM borate, pH 9.3. Peptides were separated onto a Dionex Ultimate 3000 HPLC system equipped with an AS24 strong anion exchange column, using a similar protocol to the hSAX method described previously (*Ritorto et al., 2013*). Peptides were chromatographed using a borate buffer system, namely 10 mM sodium borate, pH 9.3 (Buffer A) and 10 mM sodium borate, pH 9.3 + 0.5 M sodium chloride (Buffer B) and eluted using an exponential elution gradient into $12 \times 750$ µl fractions. The peptide fractions were desalted using SepPak-C18 SPE plates and then resuspended in 5% formic acid for LC-MS/MS analysis.

## Liquid chromatography mass spectrometry analysis (LC-MS/MS)

For tryptic digests, including tryptic + Lys-C double digests, peptide chromatography was performed using a Dionex RSLCnano HPLC. Peptides were loaded onto a 0.3 mm id × 5 mm PepMap-C18 pre-column and chromatographed on a 75 µm × 15 cm PepMap-C18 column using the following mobile phases: 2% acetonitrile +0.1% formic acid (Solvent A) and 80% acetonitrile +0.1% formic acid (Solvent B). The linear gradient began with 5% B to 35% B over 156 min with a constant flow of 300 nl/min. The peptide eluent flowed into a nanoelectrospray emitter at the front end of a Velos Orbitrap mass spectrometer (Thermo Fisher, San Jose, CA). A typical 'Top15' acquisition method was used. Briefly, the primary mass spectrometry scan (MS[1]) was performed in the Oribtrap at 60,000 resolution. Then, the top 10 most abundant m/z signals were chosen from the primary scan for collision-induced dissociation and MS[2] analysis in the Orbitrap mass analyzer at 17,500 resolution. Precursor ion charge state screening was enabled and all unassigned charge states, as well as singly charged species, were rejected.

For Lys-C digests, peptide chromatography was also performed using a Dionex RSLCnano HPLC. Peptides were directly injected onto a 75 µm × 50 cm PepMap-C18 column using the following mobile phases: 2% acetonitrile +0.1% formic acid (Solvent A) and 80% acetonitrile +0.1% formic acid (Solvent B). The linear gradient began with 5% B to 35% B over 220 min with a constant flow rate of 200 nl/min. The peptide eluent flowed into a nanoelectrospray emitter at the front end of a Q-Exactive (quadrupole Orbitrap) mass spectrometer (Thermo Fisher). A typical 'Top10' acquisition method was used. Briefly, the primary mass spectrometry scan (MS[1]) was performed in the Oribtrap at 70,000 resolution. Then, the top 10 most abundant m/z signals were chosen from the primary scan for collision-induced dissociation in the HCD cell and MS[2] analysis in the Orbitrap at 17,500 resolution. Precursor ion charge state screening was enabled and all unassigned charge states, as well as singly charged species, were rejected.

## RNA extraction and RNA-Seq

NB4 cell pellets from elutriation were resuspended in 0.25 ml PBS and immediately lysed by addition of 0.75 ml Trizol LS (Sigma, United Kingdom). RNA extraction was then performed according to manufacturer's instructions. Extracted RNA was resuspended in nuclease-free water and quantified by fluorometry using the RNA Qubit assay (Life Technologies). Fractions with similar cell cycle phase profiles were combined to produce samples enriched in G1 (fractions 1 + 2), S (fractions 3 + 4), and G2&M (fractions 5 + 6) RNA. The integrity of the total RNA was assessed using an Agilent Bioanalyser.

Two biological replicates were analyzed in technical duplicate by standard Illumina RNA-Seq. Briefly, mRNA was extracted using oligo dT beads, fragmented, then reverse transcribed using random hexamers. The cDNA was then sequenced using paired ends reads at a length of 75 bp. Each sample was run on a single lane of an Illumina HiSeq, to improve coverage of lower abundance transcripts.

A suite of custom scripts was designed to evaluate the quality of the resultant RNA-Seq data. Briefly, the data were evaluated for standard sequencing metrics including GC content and percent of reads with a quality score either greater or equal to 30. RNA-Seq specific effects were scrutinized including evenness of coverage across the transcriptome, absence of significant 3' bias, successful reduction of ribosomal RNA and high complexity of the sequenced fragments (determined by unique start and end positions of the insert).

The paired-end RNA-Seq data were then aligned to the human genome (build hg19), using TopHat, without providing a gene reference (to avoid forced mappings). Following duplicate removal using Picard's MarkDuplicate (http://picard.sourceforge.net), we quantified the gene expression of known protein coding genes using Cufflinks (*Trapnell et al., 2013*).

## siRNA depletion experiments

Depletion of Cdc20 and Cdh1/Fzr1 utilized pools of four siRNAs at a final, total concentration of 20 nM (Dharmacon). Lipofectamine RNAiMax (Life technologies) transfection reagent was used according to manufacturer's recommendations. Negative control (firefly luciferase) siRNA sequence is: 5'CGUACGC GGAAUACUUCGA. Cdc20 siRNA sequences are: 5'CGGAAGACCUGCCGUUACA, 5'GCAGAAACGG CUUCGAAAU, 5'GGGCCGAACUCCUGGCAAA, 5'GCACAGUUCGCGUUCGAGA. The lamin A/C siRNA sequence is 5'CUGGACUUCCAGAAGAACA. Cdh1/Fzr1 siRNA sequences are identical to those used in previous studies (*Emanuele et al., 2011*). Depletion of ARHGAP11A utilized pools of four siRNAs at a final, total concentration of 20 nM, unless otherwise specified (sequences: 5'UACAGACUCUUAU CGAUUA, 5'GUUCGAAGAUCUCUGCGUU, 5'GGUAUCAGUUCACAUCGAU, 5'AAGCGAUCAUUG CCAGUAG).

hTERT-RPE1 cells were harvested 24 hr post transfection for asynchronous populations. For G1 populations, hTERT-RPE1 cells were changed into serum free medium 24 hr post transfection and harvested 24 hr after that (48 hr post transfection). U2OS cells were transfected with siRNA pools targeting both Cdc20 and Cdh1 and harvested 24 hr later. Lysates were separated by SDS-PAGE, transferred to nitrocellulose membranes and immunoblotted using antibodies recognizing either GAPDH (sc-25778; Santa Cruz, USA), Cdh1 (ab3242; Abcam) or ARHGAP11A (HPA040419; Sigma UK and A303-097A; Bethyl USA) following standard procedures.

## Flow cytometry and immunoblotting of elutriated NB4 lysates

NB4 cells ($5 \times 10^5$ cells, minimum) were resuspended in cold 70% ethanol and fixed at room temperature for 30 min. The fixed cells were then washed twice with PBS and resuspended in PI stain solution (50 µg/ml propidium iodide and 100 µg/ml ribonuclease A in PBS). The cells were incubated in PI stain solution for 30 min and then analyzed by flow cytometry on a FACScalibur (BD Biosciences UK). An asynchronous population of cells was used as a control to adjust flow cytometer settings, which then remained constant throughout analysis of the set of elutriated fractions. The flow cytometry data were analyzed using FlowJo (Tree Star, Inc., OR, USA).

Lysates for SDS-PAGE analysis were prepared in lithium dodecylsulfate sample buffer (Life Technologies) and 25 mM TCEP. Samples were heated to 65°C for 5 min and then loaded onto a NuPage BisTris 4–12% gradient gel (Life Technologies), in either MOPS or MES buffer. Proteins were electrophoresed and then transferred to nitrocellulose membranes using program 3 (7 min) on the iBlot dry blotting system (Life Technologies). Membranes were then blocked in 3% milk in immunoblot wash buffer (TBS +0.1% Tween-20) for 1 hr at room temperature. Membranes were then probed with primary antibody overnight at 4°C, washed and then re-probed with HRP-conjugated secondary antibody. Primary antibodies for cell cycle immunoblot analysis were obtained from BD Biosciences (aurora kinase B), Atlas Antibodies (ARHGAP11A, HPA040830) and from Cell Signaling Technology (cyclin B1, cyclin A2, cyclin E, phospho-Histone H3-S10). Bands were visualized using enhanced chemiluminescence (Millipore Immobilon UK) and CCD camera detection (FujiFilm LAS-4000 system).

## Data analysis

The RAW data files produced by the mass spectrometer were analyzed using the quantitative proteomics software MaxQuant, version 1.3.0.5 (*Cox and Mann, 2008*). This version of MaxQuant includes an integrated search engine, Andromeda (*Cox et al., 2011*). The database supplied to the search engine for peptide identifications was a UniProt human protein database ('Human Reference Proteome' retrieved on 19 August 2012) combined with a commonly observed contaminants list. The initial mass tolerance was set to 7 p.p.m. and MS/MS mass tolerance was 0.5 Da. Enzyme was set to trypsin/P with up to 2 missed cleavages. Deamidation, oxidation of methionine and Gln->pyro-Glu were searched as variable modifications. Identification was set to a false discovery rate of 1%. To achieve reliable identifications, all proteins were accepted based on the criteria that the number of forward hits in the database was at least 100-fold higher than the number of reverse database hits, thus resulting in a false discovery rate of less than 1%. Protein isoforms and proteins that cannot be distinguished based on the peptides identified are grouped by MaxQuant and displayed on a single line with multiple UniProt identifiers. The label free quantitation (LFQ) algorithm in MaxQuant was used for protein quantitation. The algorithm has been previously described (*Luber et al., 2010*). The MaxQuant data analysis was repeated with searches for the following post-translational modifications: Phospho(STY), Methyl/Di-Methyl (KR), and Acetyl (K). Protein quantitation was performed on unmodified peptides and peptides that have modifications that are known to occur during sample processing (pyro-Glu, deamidation). All resulting MS data were integrated and managed using Data Manager, a laboratory information management system (LIMS) that is part of the PepTracker software platform (http://www.PepTracker.com).

The downstream data interpretation (protein and RNA data) of cell cycle stages in this study was performed primarily using the R language (version 0.95.262). An initial cleaning step was performed to improve the quality and value of the data set. This step involved removing proteins with less than 2 peptide identifications, those labeled as either contaminants or reverse hits and those where data were missing in any of the fractions. Proteins were further filtered using a procedure analogous to a 'checksum' function in computing. An algorithm was constructed to assess the self-consistency of the quantitation based on known relationships between the elutriated fractions. The intensities measured in the asynchronous NB4 cell population can thus be modeled as a linear combination of the intensities

originating from the six elutriated fractions that have been normalized by the measured cell count in each elutriated fraction. For each protein, the theoretical linear combination of elutriated fraction intensities (scaled by cell number) should match the protein intensity measured experimentally in the asynchronous population. Similar factors were calculated between adjacent fractions (e.g., F1 vs F2), using cell number and the proportions of cells in each phase, as determined by flow cytometry. These stringent criteria left a subset of the total proteins detected with very high data coverage across the six elutriated cell cycle fractions and high self-consistency in quantitation.

Absolute protein abundances were estimated using the iBAQ algorithm, as previously described (*Schwanhausser et al., 2011*). Gene ontology analysis was performed using the DAVID web resource (*Huang da et al., 2009*) and STRING (*Jensen et al., 2009*). Predicted transcription factor binding sites were retrieved from MSigDB (*Subramanian et al., 2005*; *Matys et al., 2006*). Tissue mRNA expression data were obtained from the Broad Cancer Map, as implemented in MSigDB (*Ramaswamy et al., 2001*; *Subramanian et al., 2005*). An arbitrary twofold cutoff was implemented to identify cell cycle varying proteins. Proteins were then clustered using the Ward algorithm into 16 clusters. 15 of these clusters were then re-clustered based on the phase of maximum expression. The final cluster, which had two peaks across the cell cycle fractions, was left unchanged (i.e., the G2&M+G1 cluster). A similar clustering analysis was performed for phosphopeptide intensities.

To carry out isoform analysis, the MaxQuant data were re-analyzed to produce isoform-specific cell cycle profiles. To do this the MS/MS information from the peptide evidences (evidence.txt), in the MaxQuant output, was used to determine the number of unique MS/MS counts in each fraction. These MS/MS counts were then averaged for peptides belonging to the same isoform, providing an isoform specific profile of MS/MS counts across fractions. This process was carried out with the additional quality filters described above, that is removal of contaminant and reverse hits and ensuring isoforms have a minimum of two unique peptide identifications. To highlight potentially interesting isoforms displaying differential behavior, a correlation score was calculated between isoforms of the same protein. Proteins showing a poor correlation between isoforms were used to identify examples of differentially regulated isoforms across the cell cycle fractions.

To compare protein and RNA data, protein identifiers were mapped to Ensembl Gene ID. Histone genes were removed from this data analysis, due to their lack of poly(A) tails. Absolute protein and mRNA abundances were plotted in DataShop, a data visualization tool developed as part of the PepTracker software suite (www.PepTracker.com). The PepTracker app runs on both Windows and Mac OSX and is freely available for download (www.PepTracker.com/ds/).

## Data sharing

Gene expression data sets are provided in multiple forms to facilitate access for a range of end-users. MS raw files can be accessed from the EBI PRIDE database (accession PXD000678). RNA-Seq raw files will be available from EBI (accession EGAD00001000736). Peptide evidence data derived from MaxQuant have been deposited to Dryad and can be accessed using this hyperlink: http://dx.doi.org/10.5061/dryad.2r79qL (*Ly et al., 2014*). Protein, phosphoprotein, and RNA identifications and quantitations are available in supplementary tables to this manuscript (e.g., *Supplementary files 1, 4 and 6*). Outputs from the cell cycle gene expression analysis, cell line meta-analysis, and comparative protein and mRNA analysis are also provided in supplementary tables (*Supplementary files 2, 3, and 5*).

In addition, gene-by-gene visualization of the quality-filtered data set of protein and mRNA expression for over 6000 genes analyzed across the cell cycle is provided in a searchable, online format via the Encyclopedia of Proteome Dynamics (EPD) (http://www.peptracker.com/epd) (*Larance et al., 2013*). This is a web-based tool, part of the PepTracker platform, which aims to visually communicate and disseminate data from large scale, multi-dimensional proteomic experiments. The EPD is developed using Python and the Django web framework. The EPD uses an Oracle database to store raw data, including the protein and mRNA data from this study. The visualizations depicting protein and RNA data are created using the R programming language and integrated into the web tool via the RPy2 library.

## Acknowledgements

We thank our colleagues in the Lamond group for advice and discussion and in particular, we thank Dalila Bensaddek, Aki Endo, Saskia Hutten, Mark Larance, and Armel Nicolas for providing feedback on the manuscript, Vackar Afzal for help with PT Data Manager and Robert Kent for help with PT Datashop. We thank former lab members Francois-Michel Boisvert and Yun Wah Lam for their contributions to

the origins of the project. We thank Rosie Clarke (Centre for Advanced Scientific Technologies, University of Dundee) for advice on flow cytometry analysis. We thank Matthias Trost and Maria Stella Ritorto (MRC Protein Phosphorylation and Ubiquitylation Unit, University of Dundee) for advice on peptide fractionation using hSAX. AIL is a Wellcome Trust Principal Research Fellow.

## Additional information

### Funding

| Funder | Grant reference number | Author |
| --- | --- | --- |
| Wellcome Trust | 083524/Z/07/Z, 097945/B/11/Z, 073980/Z/03/Z, 081361/Z/03/Z | Tony Ly, Yasmeen Ahmad, Angus I Lamond |
| European Union FP7 Prospects Network | HEALTH-F4-2008-201648 | Tony Ly, Yasmeen Ahmad, Angus I Lamond |
| Biotechnology and Biological Sciences Research Council | BB/K003801/1 | Angus I Lamond |
| Wellcome Trust | 098051 | Adam Shlien, Michael R Stratton |
| EpiGeneSys Network | HEALTH-F4-2010-257082 | Tony Ly, Yasmeen Ahmad, Angus I Lamond |

The funders had no role in study design, data collection and interpretation, or the decision to submit the work for publication.

### Author contributions

TL, Designed and performed most of the experiments, Analyzed the RNA-Seq and proteomics data, Wrote the manuscript; YA, Analyzed the RNA-Seq and proteomics data, Performed the isoform data analysis, Helped write the manuscript; AS, Acquired and analyzed the RNA-Seq data, Helped write the manuscript; DS, AM, Acquired and analyzed the data shown in *Figures 10C,D*; MJE, Designed and mentored the experiments resulting in the data shown in *Figures 10C,D*, Helped write the manuscript; MRS, Helped conceive and design the project, Helped acquire the RNA-Seq data; AIL, Conceived, designed, and mentored the project, Analyzed the proteomics and RNA-Seq data, Wrote the manuscript

## Additional files

### Supplementary files

• Supplementary file 1. Proteomics data set. This file summarizes the proteins identified and quantified in asynchronous NB4 cells and in the fractions produced by elutriation, and includes the following data for each protein identification: protein and gene identifiers, protein descriptions, sequence coverage, the number of supporting peptides, the posterior error probabilities (PEPs), the extracted ion chromatogram (XIC) intensities, the LFQ-normalized intensities, the iBAQ-scaled intensities, and mapped transcript FPKM values from the RNA-Seq data.

• Supplementary file 2. Cell line proteome meta-analysis. Comparison of the proteomic data set obtained for NB4, a human promyelocytic leukemia cell line that grows in suspension culture, with other recent examples of in depth proteomic analysis of different human cell lines, most of which are adherent tumor cell lines of either fibroblast or epithelial origin. In total, the meta-analysis included protein data from 14 cell line proteomes: 3 × HeLa, 2 × U2OS, A549, GAMG, HEK293, K562, LnCap, MCF7, RKO, HepG2, and Jurkat-T (*Lundberg et al., 2010*; *Beck et al., 2011*; *Nagaraj et al., 2011*; *Geiger et al., 2012*), which were consolidated and mapped to Ensembl Genes prior to comparison. The combined data set provides evidence of protein-level expression of over 11,000 genes. Of these, a common set of ~3000 genes are identified by protein data from all these cell lines, defining a core, shared proteome (Columns D and E), and >1000 genes are uniquely detected in this analysis of NB4 cells (Columns A and B). A focused comparison of NB4, K562, Jurkat-T, HeLa and MCF7 cell line

proteomes reveals ~90 genes that are specifically expressed in myeloid cell lines NB4 and K562 (Columns G and H).

• Supplementary file 3. Proteins whose Abundance is cell cycle regulated. For quantitation, the proteomic data set was filtered to only include proteins that were detected in asynchronous cells and all six elutriation fractions. Of these ~6500 proteins, 358 (~5.5%) are proteins whose abundance is cell cycle regulated (i.e., varies in abundance by at least two-fold across the fractions). These proteins vary in expression profile, and cluster into seven distinct groups that differ primarily in peak fraction. Gene and protein identifiers, cluster membership, and motifs that are predicted to modulate post-translational regulation are provided below. Other than the Dbox (R-x-x-L from King et. al, Mol. Biol. Cell 1996, 7, 1343-1357), motif sequences were obtained from the Eukaryotic Linear Motif resource (ELM).

• Supplementary file 4. Phosphopeptide dataset. This file summarizes the ~2700 phosphopeptides identified and quantified in asynchronous NB4 cells and in the fractions produced by elutriation, and includes the following data for each phosphopeptide identification: protein and gene identifiers, protein descriptions, the phosphopeptide sequence, localization scores and probabilities, posterior error probabilities (PEPs), the Andromeda search scores, the mass error, and the extracted ion chromatogram (XIC) intensity.

• Supplementary file 5. Proteins whose phosphorylation is cell cycle regulated. This file summarizes the cell cycle varying phosphopeptides that were identified without phospho-specific enrichment. These phosphosites were filtered to only include phosphopeptides that were independently identified in asynchronous cells and in all elutriation fractions. A minor fraction of these phosphopeptides (89 phosphopeptides, or 3% of the total phosphopeptides identified in this data set, corresponding to 79 phosphoproteins) vary by at least two-fold across the elutriation fractions. Cell cycle regulated phosphopeptides are listed below with Andromeda database search scores, localization probabilities, posterior error probabilities (PEPs), and intensities in each fraction.

• Supplementary file 6. RNA-Seq data set. This file provides gene identifiers, counts, and data quality markers for protein coding genes identified in any of the elutriated samples. The six elutriated fractions were pooled into three samples (F1+F2, F3+F4, F5+F6). mRNA was then separately extracted from these pooled samples using oligo dT beads, fragmented, then reverse transcribed using random hexamers. The cDNA was then sequenced using paired ends reads at a length of 75 bp. Each sample was run on a single lane of an Illumina HiSeq, to improve coverage of lower abundance transcripts. The paired-end RNA-Seq data were then aligned to the human genome (build hg19), using TopHat, without providing a gene reference (to avoid forced mappings). Following duplicate removal using Picard's MarkDuplicate (http://picard.sourceforge.net), we quantified the gene expression of known protein coding genes using Cufflinks (*Trapnell et al., 2013*). Genes with low data quality were removed from subsequent data analysis.

### Major datasets

The following datasets were generated:

| Author(s) | Year | Dataset title | Dataset ID and/or URL | Database, license, and accessibility information |
|---|---|---|---|---|
| Ly T, Ahmad Y, Shlien A, Soroka D, Mills A, Emanuele MJ, Stratton MR, Lamond AI | 2014 | Peptide Evidence | http://dx.doi.org/ 10.5061/dryad.2r79q | Publicly available at Dryad (http://datadryad.org/). This compressed file contains a tab-delimited table listing the peptide evidence generated by processing the raw MS and MS/MS data using the MaxQuant software package, which includes a built-in database search engine called Andromeda. The spectra were searched against the UniProt Human Reference Proteome, accessed on August 2012. The table includes all instances of peptide identifications and quantitations, and their database search scores and posterior error probabilities. |

| Ly T, Ahmad Y, Shlien A, Soroka D, Mills A, Emanuele MJ, Stratton MR, Lamond AI | 2014 | Raw Mass Spectra | PX000678; http://www.ebi.ac.uk/pride/archive/projects/PXD000678 | Publicly available at the EBI PRIDE database (http://www.ebi.ac.uk/pride/archive/). |
| Ly T, Ahmad Y, Shlien A, Soroka D, Mills A, Emanuele MJ, Stratton MR, Lamond AI | 2014 | RNA-Sequencing Reads | EGAD00001000736; https://www.ebi.ac.uk/ega/datasets/EGAD00001000736 | Publicly available at the EBI European Genome-phenome Archive (https://www.ebi.ac.uk/ega/home). |

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
