## [Decision Letter]

Thank you for sending your work entitled “A Quantitative Chronology of Gene Expression through the Cell Cycle in Human Myeloid Leukemia Cells” for consideration at *eLife*. Your article has been favorably evaluated by a Senior editor and 3 reviewers, one of whom is a member of our Board of Reviewing Editors.

The Reviewing editor and the other reviewers discussed their comments before we reached this decision, and the Reviewing editor has assembled the following comments to help you prepare a revised submission.

In this study the authors have undertaken a comprehensive analysis of the transcriptome and proteome of a human myeloid leukaemia cell line through the cell cycle. The authors have used elutriation to enrich for specific cell cycle phases thereby obviating the need to activate cell cycle checkpoints to synchronise cells. The authors make a number of observations, some of which are novel and others generalize principles that have already been observed in the characterization of individual cell cycle regulators. The mass spectrometry analysis is very comprehensive and represents probably the most thorough analysis of protein levels through the cell cycle to date. In addition, this is the first study to compare the mRNA to protein levels through the cell cycle and they find, as largely expected from what we know of protein regulation by proteolysis in the cell cycle, that mRNA levels poorly predict protein levels.

There are two main categories of concern. The first is the methodology used in the paper, which should be clarified according to the comments below. The second is the analysis of the results, which requires very substantial revision. The paper has the peculiar appearance of being written in isolation from the cell cycle literature. A number of the proteins selected as examples by the authors have been extensively analysed in other studies, but these are not referenced. For example, the authors focus on cyclin A2 and by comparing its mRNA level to protein level they conclude that transcription is a major factor in regulating cyclin A levels. They make no mention of the considerable work that has been done on cyclin A in human cells, which has shown that it is an E2F target whose transcription starts at the end of G1 phase, and is targeted by the APC/C ubiquitin ligase in early mitosis and G1 phase. Thus, it is the combination of transcriptional control and cell cycle-dependent proteolysis that controls cyclin A levels. The authors are similarly naïve in their conclusion that protein levels are poorly correlated with mRNA levels; the extensive studies in the last 20 years on SCF-dependent and APC/C-dependent proteolysis is basically ignored with the primary references here being papers from the mid 1990's. Similarly, the authors appear not to have realized that because ∼50% of the population of asynchronous cells are in G1 phase, mRNA levels in asynchronous cells will be poorly predictive of protein level for any protein whose transcription or stability is cell cycle regulated.

Technical issues:

The analysis of single shot proteome and deep proteome have been merged; this makes it difficult to extract the depth of each analysis and information on replicates. How many replicates are there for the deep proteome? This would mean information on abundance, splicing and phosphorylation derived from this dataset are based only on one sample? The data are therefore of limited quantitative utility but provide a through qualitative description.

It is not clear how the cell cycle proteomes collected; are they deep or single shot? How many replicates? The comparison is limited to proteins that are identified in all fractions, however this would mean that proteins that would be specifically induced or restricted expression to defined cell cycle phases would not be part of the analysis? Only proteins whose expression oscillates will be detected.

Specific points:

1) I would suggest adding short descriptions to the data in the supplementary spreadsheets. This will allow the reader to keep a better overview over the provided material.

2) [Supplementary-material SD2-data]: it would be useful to also provide protein names.

3) Figure 2: If available, it would be informative to also see the <2N events in the flow cytometry cell cycle profiles.

4) Table 1: It is surprising that STAT3 annotation does not appear in G1 and S phase.

5) Figure 5: Isn't the cluster 6 in the 3rd row, 2nd column supposed to be G2/M? The same applies to Figure 5, especially since the authors use the expression “G2/M” in the figure legend. Also, I would label the panels in Figure 5 with the respective elutriation fractions. Furthermore, “complex” to me is “M/G1”. 5b cluster analysis is very hard to see.

6) Figure 6: It would be interesting to know how large the dataset on splicing isoforms is. The protein coverage and location of discriminatory peptides should be highlighted. It is not clear if the identification of these peptides as PEPs are significant or if based on FDR could these be coincidence. The data for PPFIBP1 and SDCCAG8 are not convincing since the differences between several of the isoforms appear very small but the effect much larger. Inclusion of supporting material such as extracted intensities for the isoform specific peptides will help convince validity of these observations. These are from only 1 analysis and there is no validation. How many protein isoforms are detected that fulfil the 2 peptide criteria? Is there any correlation with transcript information?

7) Section on phospho-proteomics/[Supplementary-material SD5-data]: the only detected Tyr phosphorylation is mapped to Cdk2; however, the protein identifier (P06493) suggests Cdk1.

8) It should be mentioned that ARHGAP11A is predicted to encode a RhoGAP.

9) Figure 10: Blots should be re-probed with e.g. anti-phospho-Histone H3 (Ser10) to exclude cell cycle synchronization effects of Cdh1/Cdc20 depletion that could indirectly result in elevated ARHGAP11A levels and a probing with anti-cyclinB could show APC/C inactivation. The serum-starved RPE-1 cells shown in would help to exclude this, however the probing is barely visible. I would suggest splitting the panel and increasing contrast or reprobe the blot with longer exposure.

10) Also, I would have preferred to see an ARHGAP11A depletion to validate the antibody. Given that the data presented in Figure 10 are hardly discussed in the text, I would say this could be omitted.

11) In Figures 6 and 7, the results on splice isoform and phospho-peptides abundances are presented, demonstrating that both relative isoform abundance and phospho-site occupancy vary for some proteins over the cell cycle. It could be that it has escaped my attention, but it was not clear to me how extensive the data set on splice isoform abundances is – this information would be of great value for the interpretation of the proteome data. Furthermore, due to the lack of enrichment, the yield of phospho-peptides is rather low and the fraction of cell cycle-regulated phospho-peptides appears quite small (3.4%). I would thus say that these two figures are not adding any substantial value to the publication.

12) The paper is very detailed (12 main figures) and could benefit from shortening; several figures together with technical and standardising information should be placed in supplementary material.

---

## [Author Response]

*There are two main categories of concern. The first is the methodology used in the paper, which should be clarified according to the comments below. The second is the analysis of the results, which requires very substantial revision. The paper has the peculiar appearance of being written in isolation from the cell cycle literature. A number of the proteins selected as examples by the authors have been extensively analysed in other studies, but these are not referenced. For example, the authors focus on cyclin A2 and by comparing its mRNA level to protein level they conclude that transcription is a major factor in regulating cyclin A levels. They make no mention of the considerable work that has been done on cyclin A in human cells, which has shown that it is an E2F target whose transcription starts at the end of G1 phase, and is targeted by the APC/C ubiquitin ligase in early mitosis and G1 phase. Thus, it is the combination of transcriptional control and cell cycle-dependent proteolysis that controls cyclin A levels*.

We welcome the opportunity to relate more accurately the analysis of our dataset to extant literature in the cell cycle field. In particular, we have now included in the revised manuscript references to recent work documenting the mechanisms underlying the cell cycle regulation of cyclin A, including transcriptional activation by E2F in S phase and targeted degradation by the APC/C in G1 phase.

*The authors are similarly naïve in their conclusion that protein levels are poorly correlated with mRNA levels; the extensive studies in the last 20 years on SCF-dependent and APC/C-dependent proteolysis is basically ignored with the primary references here being papers from the mid 1990's. Similarly, the authors appear not to have realized that because ∼50% of the population of asynchronous cells are in G1 phase, mRNA levels in asynchronous cells will be poorly predictive of protein level for any protein whose transcription or stability is cell cycle regulated*.

We respectfully disagree that we have been ‘*naïve*' in our conclusion that, in general, protein levels are poorly correlated with mRNA levels. As this study, so far as we know, provides the most comprehensive global analysis of cell cycle regulated gene expression to date, at both the protein and mRNA level (and the only one we know of in myeloid cells), our finding that the relationship between protein and mRNA levels is highly complex and gene‐specific is novel. We are not aware of any comparable, prior analyses that would allow this conclusion to be drawn. While it is true that for certain specific genes, as previously described, protein and mRNA levels are highly correlated due to synergistic transcriptional activation and targeted degradation (such as cyclin A), we specifically show that there are many other cell cycle regulated genes where protein and mRNA levels are very poorly correlated and in some cases even anti‐correlated. It is therefore not the case that the prior literature had already comprehensively described how cell cycle affects gene expression at the protein and mRNA level and also not the case that all, or even most, cell cycle regulated genes are regulated in the same way. This in itself we feel is a rather interesting result.

To clarify this point further, we would stress that although our data show that overall the protein and mRNA abundances of cell cycle regulated genes are less correlated in asynchronous cells, we also find that there is a subset of cell cycle regulated genes (most of which peak in G2/M) whose protein and mRNA abundances in asynchronous cells are actually more correlated (r = 0.76) than bulk gene expression (r = 0.63). These data therefore do not fit with the statement from the reviewers that “mRNA levels in asynchronous cells will be poorly predictive of protein level for any protein whose transcription or stability is cell cycle regulated”. The clear implication of this statement is that cell cycle regulated genes are either modulated by transcriptional, or stability, mechanisms and therefore it should be apparent that they will inevitably show a poorer than average correlation between protein and mRNA levels. Instead, however, our data show that for certain cell cycle regulated genes, mRNA and protein levels are more correlated in asynchronous cells than seen for bulk gene expression. We hope that these results will thus be of general interest to the cell biology community and can serve to stimulate further research into the mechanisms of gene expression and how it contributes to cell cycle regulation.

*The analysis of single shot proteome and deep proteome have been merged; this makes it difficult to extract the depth of each analysis and information on replicates. How many replicates are there for the deep proteome? This would mean information on abundance, splicing and phosphorylation derived from this dataset are based only on one sample? The data are therefore of limited quantitative utility but provide a through qualitative description*.

*It is not clear how the cell cycle proteomes collected; are they deep or single shot? How many replicates*?

We have now revised the manuscript to clarify these technical details of our experiment, i.e., that three biological and three technical replicates were performed with the single shot proteome analysis, while the deep proteome analysis represents one large amalgamated biological experiment. While we of course agree that in general more replicate data sets enhance the estimation of variance, we disagree with the notion that whether data are deemed either quantitative or qualitative is simply a reflection of the number of replicates performed. For example, a poorly designed experiment, with low quality data, would not be very ‘quantitative’, even with 3 or more replicates! We note that in our deep proteome analysis, even although it corresponds to one very large amalgamated experiment (and can thus be viewed as a single replicate), multiple internal replications help to mitigate noise in the data and to improve the accuracy of quantitation. This includes the fact that the separate proteomic measurements of the different elutriated fractions actually represent multiple technical replicates within the experiment. Furthermore, the filtered dataset we used to identify cell cycle regulated proteins had, on average, 22 independent, quantitative peptide measurements to support the final quantitation value presented for each protein. Thus, we strongly assert that this study is, by any reasonable definition, “quantitative” and allows quantitative conclusions to be drawn, albeit with the usual limitations affecting all quantitative studies. In this case the scale of the deep proteome experiment and huge volume of instrument and analysis time involved simply precludes it being done with many biological replicates, given current technology. However, we do not wish this partly semantic point as to how quantitative the data are to be contentious. Therefore, as a compromise, we changed the title from “A quantitative chronology...” to “A proteomic chronology....”, while retaining the descriptor “quantitative” in the abstract and the text. We hope this will be acceptable.

*The comparison is limited to proteins that are identified in all fractions, however this would mean that proteins that would be specifically induced or restricted expression to defined cell cycle phases would not be part of the analysis? Only proteins whose expression oscillates will be detected*.

We agree that the comparison here is limited to proteins that are identified in all elutriated fractions. However, we believe the concern raised by the reviewers is not in fact valid because the elutriated fractions are only enriched for cells at different cell cycle stages and not completely pure, (as shown in Figure 2). Therefore, our analysis does not necessarily exclude detection of proteins that are specifically induced only at defined cell cycle stages. Even if their regulation was so strict that they were completely absent in one or more stages, rather than simply modulated in abundance, they should likely still be present at some level and hence in theory be detectable in each of the elutriated fractions because these contain mixed populations. Our decision to enforce only considering proteins where data were detected across all fractions is deliberately conservative and we accept this reduces the total number of proteins we evaluate for cell cycle regulation. On the positive side, this decision to be conservative improves the overall quality and quantitative value (see comments also above) of our data set, reduces false positives and ensures that we include predominantly proteins that are comfortably above the sensitivity threshold for our technical workflow.

*1) I would suggest adding short descriptions to the data in the supplementary spreadsheets. This will allow the reader to keep a better overview over the provided material*.

Full descriptions of the data contained within the supplementary spreadsheets have now been added to the header of each spreadsheet as requested. The headers also include information about how the data were generated and column descriptions.

*2)*
[Supplementary-material SD2-data]*: it would be useful to also provide protein names*.

Gene names have now been added to aid data interpretation. [Supplementary-material SD2-data] contains the results of the cell line meta‐analysis, which was performed at the gene, rather than protein, level. Thus, we felt it would be more appropriate to provide gene names.

*3)*
Figure 2*: If available, it would be informative to also see the <2N events in the flow cytometry cell cycle profiles*.

The cell cycle distributions shown includes cells with <2N DNA content, which were infrequently observed. We have added a statement to the Figure 2 caption to clarify this point. The only gates employed were along the “width” dimension, which were designed specifically to remove either cell clumps, or cell doublets.

*4)*
Table 1*: It is surprising that STAT3 annotation does not appear in G1 and S phase*.

This is an interesting point. What we find is that out of 56 G1‐peaking proteins, 22 have predicted STAT3 binding sites in their promoters. However, due to the prevalence of STAT3 binding sites in the genome, the resulting p‐value for STAT3 enrichment is higher than our threshold p‐value of 0.05. Thus, while STAT3 is likely to be important for regulating specific genes, our data suggest that on the whole, other transcription factors (such as E2F in S‐phase), are predominantly responsible for general transcriptional regulation of cell cycle gene expression in G1‐ and S‐phase. However, STAT3 appears to be generally important in regulating the transcription of proteins that peak in G2/M/G1, with nearly all members of this cluster having predicted STAT3 binding sites in their promoters. We welcome the opportunity to discuss these interesting findings, which highlight the potential value of conducting a system‐wide, unbiased analysis.

*5)*
Figure 5*: Isn't the cluster 6 in the 3rd row, 2nd column supposed to be G2/M? The same applies to*
Figure 5*, especially since the authors use the expression “G2/M” in the figure legend. Also, I would label the panels in*
Figure 5
*with the respective elutriation fractions. Furthermore, “complex” to me is “M/G1”. 5b cluster analysis is very hard to see*.

We have changed all instances of “G2” to “G2/M”. We agree that the label “complex” is ambiguous, and should be renamed. However, due to the overlapping size distributions of G2‐ and M‐ phase cells, we cannot separate these two phases by elutriation and thus we feel that the label “M/G1” would be inappropriate. Therefore, we have renamed “complex” with “G2/M/G1”. All cluster labels have also been enlarged to improve clarity.

*6)*
Figure 6*: It would be interesting to know how large the dataset on splicing isoforms is*.

Our intention is to highlight examples of proteins where the peptide coverage is of sufficient depth and quality to enable discrimination of potential isoform‐specific, cell cycle behavior. To our knowledge, this study is the first to demonstrate that isoform‐specific cell cycle expression is detectable using a large‐scale proteomics approach. This was due, at least in part, to the high average peptide sequence coverage achieved, an important point that we are keen to emphasise. We also note, however, that a comprehensive analysis of isoform‐specific variation is still very difficult, if not impossible, with current technology, given that even the most detailed proteomic datasets provide at best ∼40% mean sequence coverage. This important point is now mentioned explicitly in the revised results section.

*The protein coverage and location of discriminatory peptides should be highlighted. It is not clear if the identification of these peptides as PEPs are significant or if based on FDR could these be coincidence. The data for PPFIBP1 and SDCCAG8 are not convincing since the differences between several of the isoforms appear very small but the effect much larger. Inclusion of supporting material such as extracted intensities for the isoform specific peptides will help convince validity of these observations*.

We have revised and improved Figure 6 by annotating the locations where peptide evidence was detected within the sequence diagrams. Furthermore, we have changed the color scheme to help readers distinguish isoform expression patterns, and relate the expression patterns to the sequence diagrams. The full evidence table, which includes extracted ion intensities, posterior error probabilities (PEPs), and Andromeda database search scores for all peptide evidences, is now provided as supplementary material (Dryad DOI).

*These are from only 1 analysis and there is no validation. How many protein isoforms are detected that fulfil the 2 peptide criteria*?

All three examples highlighted fulfill the 2 peptide criterion; that is, each protein isoform is supported by at least two peptides with unique sequences.

*Is there any correlation with transcript information*?

We agree that a comparison of protein and transcript isoform data would be highly interesting, and we are actively pursuing this analysis. However, comparing isoform‐specific protein and mRNA data is challenging due to mapping inconsistencies (e.g., Ensembl genomic annotations vs UniProt), and would be significantly improved with increased sequence coverage. We hope to resolve these challenges and provide a comprehensive analysis in the future.

*7) Section on phospho-proteomics/*[Supplementary-material SD5-data]*: the only detected Tyr phosphorylation is mapped to Cdk2; however, the protein identifier (P06493) suggests Cdk1*.

The N‐terminal sequence of Cdk1 and Cdk2 are highly sequence similar. Proteolytic digestion of either protein produces Tyr15‐containing peptides that have identical sequence. Thus, we cannot uniquely assign this phosphorylation site to either protein. The wording in the text has been changed to reflect this ambiguity.

*8) It should be mentioned that ARHGAP11A is predicted to encode a RhoGAP*.

Agreed – this is now mentioned in the text. Furthermore, we have added a reference to a recently published study on ARHGAP11A function suggesting that the protein is important in cytokinetic furrow specification.

*9)*
Figure 10*: Blots should be re-probed with e.g. anti-phospho-Histone H3 (Ser10) to exclude cell cycle synchronization effects of Cdh1/Cdc20 depletion that could indirectly result in elevated ARHGAP11A levels and a probing with anti-cyclinB could show APC/C inactivation. The serum-starved RPE-1 cells shown in*
Figure 10
*would help to exclude this, however the probing is barely visible. I would suggest splitting the panel and increasing contrast or reprobe the blot with longer exposure*.

As requested, we have replaced the blots to make clearly visible the stabilization of ARHGAP11A protein in serum‐starved, Cdh1‐depleted RPE‐1 cells, which helps exclude the possibility that elevated ARHGAP11A levels are due to a cell cycle synchronization artefact. Furthermore, the blots have been probed with anti‐cyclin B1 to show APC/C inactivation.

*10) Also, I would have preferred to see an ARHGAP11A depletion to validate the antibody. Given that the data presented in*
Figure 10
*are hardly discussed in the text, I would say this could be omitted*.

We had already performed ARHGAP11A depletion experiments to show the specificity of both antibodies used. We have therefore added these data, as requested, to the supporting material (Figure 10—figure supplement 1).

*11) In*
Figures 6 and 7*, the results on splice isoform and phospho-peptides abundances are presented, demonstrating that both relative isoform abundance and phospho-site occupancy vary for some proteins over the cell cycle. It could be that it has escaped my attention, but it was not clear to me how extensive the data set on splice isoform abundances is – this information would be of great value for the interpretation of the proteome data. Furthermore, due to the lack of enrichment, the yield of phospho-peptides is rather low and the fraction of cell cycle-regulated phospho-peptides appears quite small (3.4%). I would thus say that these two figures are not adding any substantial value to the publication*.

We would respectfully argue that the figures and the data contained therein are of substantial value to the cell cycle community, as this study is the first to document phosphorylation in a large‐scale manner in a minimally perturbed mammalian system, and the first to examine the cell cycle gene expression in this cell lineage. We agree that a comprehensive analysis of phosphorylation in this minimally perturbed system using phospho‐enrichment would also be of interest to the community, and we are actively pursuing this. However, even without phospho‐enrichment, we still detect here over 2000 phosphorylation sites, which is not insubstantial given that recent phospho‐enrichment studies typically report ∼10,000 phosphorylation sites. Moreover, this includes many phosphorylation sites that are known to play important regulatory roles in the cell cycle (thus validating the methodology), as well as many novel phosphorylation sites. In other words, although we agree that due to lack of phospho‐ enrichment, the total number of phosphorylation sites reported here is lower than in some previous studies specifically addressing this issue, we strongly argue that the phosphorylation data presented here have considerable biological value. Furthermore, since the overall high depth of peptide sequence obtained here was based upon including also the phospho‐peptides, we feel it is hard to argue that deliberately excluding these data can be helpful.

*12) The paper is very detailed (12 main figures) and could benefit from shortening; several figures together with technical and standardising information should be placed in supplementary material*.

We appreciate this study is unusually detailed. To a large extent this is an inevitable corollary of aiming to document global gene expression dynamics across such a complex biological process as the cell cycle. We are especially pleased that *eLife* provides a unique opportunity to include a comprehensive treatment of the data that would definitely not be possible in the standard high profile journals. To keep the length of the manuscript within reasonable limits, we have however revised the manuscript to try to be as concise as possible, commensurate with the scale and scope of the dataset.

While this manuscript is somewhat long and detailed, the overall length may have been overestimated during review because the original submission file included interdigitated supplementary figures amongst the main figures. We understood this to be the desired layout in an *eLife* publication. These figures, labelled “Figure X–supplementary figure Y”, are to be hyperlinked from the main figure when viewed online, thus providing readers convenient access to supplementary figures. We apologize that this was not made clear in the original file containing the figures and wonder if this contributed to giving the impression that the manuscript was longer than it will actually appear.